



Earth System Science Data Discussions — Open Access

# LGHAP v2: A global gap-free aerosol optical depth and PM$_{2.5}$
# concentration dataset since 2000 derived via big earth data analytics
Kaixu Bai[1,2], Ke Li[1], Liuqing Shao[1], Xinran Li[1], Chaoshun Liu[1], Zhengqiang Li[3], Mingliang Ma[4],
Di Han[1], Yibing Sun[1], Zhe Zheng[1], Ruijie Li[1], Ni-Bin Chang[5], Jianping Guo[6]
[1]Key Laboratory of Geographic Information Science (Ministry of Education), School of Geographic Sciences, East China
Normal University, Shanghai 200241, China
[2]Institute of Eco-Chongming, 20 Cuiniao Rd., Chongming, Shanghai 202162, China
[3]State Environmental Protection Key Laboratory of Satellite Remote Sensing, Aerospace Information Research Institute,
Chinese Academy of Sciences, Beijing 100101, China
[4]School of Surveying and Geo-Informatics, Shandong Jianzhu University, Jinan 250101, China
[5]Department of Civil, Environmental, and Construction Engineering, University of Central Florida, Orlando, FL, USA
[6]State Key Laboratory of Severe Weather, Chinese Academy of Meteorological Sciences, Beijing, China
*Correspondence to*: Kaixu Bai (kxbai@geo.ecnu.edu.cn) and Jianping Guo (jpguocams@gmail.com)
**Abstract.** The Long-term Gap-free High-resolution Air Pollutants concentration dataset (LGHAP) provides spatially
contiguous daily aerosol optical depth (AOD) and particulate matters (PMs) concentration at 1-km grid resolution in China
since 2000. This advancement empowered some unprecedented assessments of aerosol variations and its impacts on
environment, health, and climate in the past few years. However, there is a need to improve such a MODIS-like gap-free high
resolution AOD and PM$_{2.5}$ concentration dataset with new robust features. In this study, we present the version 2 of such a
global-scale LGHAP dataset (LGHAP v2) that was generated using an improved big earth data analytics approach via a
seamless integration of distinct data science, pattern recognition, and deep learning methods. To better reconstruct global AOD
distribution from daily MODIS AOD imageries, multimodal AODs and air quality measurements acquired from relevant
satellites, ground monitoring stations, and numerical models across the globe throughout the past two decades were firstly
harmonized by harnessing the capability of random forest-based data-driven models. Then, an improved tensor-flow-based
AOD reconstruction algorithm was developed to weave harmonized multi-source AODs products together for gap-filling. The
results of ablation experiments demonstrated the improved tensor-flow-based gap filling method has a better performance in
terms of both convergence speed and data accuracy. Ground-based validation results indicated a good data accuracy of the
global gap-filled AOD dataset, with R of 0.85 and RMSE of 0.14 compared against worldwide AOD observations from
AERONET, which is better than the purely reconstructed AODs (R=0.83, RMSE=0.15) and slightly worse than raw MAIAC
AOD retrievals from Terra (R=0.88, RMSE=0.11). A novel deep learning model, named as the scene-aware ensemble learning
graph attention network (SCAGAT), was developed to better predict PM$_{2.5}$ concentrations across the globe. By gaining better
spatial representativeness of data-driven models across regions, the SCAGAT algorithm performed better during spatial
extrapolation, largely reducing modeling biases over regions even though in situ PM$_{2.5}$ concentration measurements are limited
or absent. Site-specific validation results indicated that the gap-free PM$_{2.5}$ concentration estimates exhibit higher prediction
accuracies with R of 0.95 and RMSE of 5.7 µg m$^{-3}$, compared against the PM$_{2.5}$ concentration measurements obtained from
priorly held-out sites worldwide. Overall, leveraging state-of-the-art methods in data science and artificial intelligence, a
quality-enhanced LGHAP v2 dataset was generated through big earth data analytics by weaving multimodal AODs and air
quality measurements from different sources together cohesively. The gap-free, high-resolution, and global coverage merits
render LGHAP v2 dataset an invaluable data base to advance aerosol- and haze-related studies and trigger multidisciplinary
applications for environmental management, health risk assessment, and climate change analysis. All gap-free AOD and PM$_{2.5}$
grids in the LGHAP v2 dataset are shared online publicly (Bai et al., 2023a), with data user guide and relevant visualization
codes available at https://doi.org/10.5281/zenodo.10216396.

## 1 Introduction

Atmospheric aerosols, either natural or anthropogenic, have been proven to pose significant threats to human health, ambient environment, and climate (Up in the aerosol, 2022). The risks to public health from aerosol pollution are clear, with about 4.2 million deaths per year attributable to the exposure of fine aerosol particles, as stated by the World Health Organization (WHO, 2022). With increased aerosol loading, aerosols can significantly impair atmospheric visibility due to the hygroscopic effect, thereby reducing direct solar radiation on the Earth's surface (Liu et al., 2020; Wang and Yang, 2014; Wild et al., 2021; Yang et al., 2016). In addition to evident impacts on air quality (Li et al., 2017), atmospheric aerosols also have an important and complex influence on regional and even global climate (Anon, 2022; Guo et al., 2016, 2019; Li et al., 2019; Yang et al., 2020; Zhao et al., 2020). Therefore, an accurate monitoring of atmospheric aerosol loading is vital for improving our understanding of human-driven ambient environment and exposure pathways in health risk assessment.

Aerosol optical depth (AOD), a measure of aerosols distributed within an air column from the Earth's surface to the top of the atmosphere, has been widely used as a key indicator of total atmospheric aerosol loading. AOD observations from ground monitoring stations have long been recognized as the ground truth, and a few ground-based aerosol observing networks, e.g., the internationally collaborated Aerosol Robotic Network (AERONET), China Aerosol Remote Sensing Network (CARSNET), and Sun−Sky Radiometer Observation Network (SONET), had been established to provide global and/or regional aerosol measurements (Che et al., 2015; Giles et al., 2019; Li et al., 2018). However, the sparse distribution of ground monitoring stations poses significant challenges to gain a better understanding of aerosol variations across the globe.

Satellite-based AOD products well bridge such a gap by providing spatially-resolved AOD retrievals with a vast spatial coverage. A variety of space-borne instruments, e.g., Sea-viewing Wide Field-of-view Sensor (SeaWiFS), Moderate Resolution Imaging Spectroradiometer (MODIS), Visible Infrared Imaging Radiometer Suite (VIIRS), and Polarization and Directionality of the Earth's Reflectances (POLDER), had been deployed onboard different satellite platforms and launched into space over the past forty years (Wei et al., 2020).These versatile instruments provide ample AOD and aerosol measurements, enabling us to map global AOD distribution with finer spatial resolutions in a long run. Nonetheless, satellite-based AOD retrievals often suffer from excessive data gaps due to extensive cloud covers and retrieval failures, significantly impairing the application potential of these spatially incomplete AOD imageries. Moreover, substantial data gaps in satellite-based AOD products could result in large uncertainties when assessing aerosol impacts on weather and climate.

A variety of gap-filling methods were developed and applied to reconstruct missing values in satellite remotely sensed AOD images (Wei et al., 2020; Xiao et al., 2021). The simplest method is to fill in data gaps with valid observations from other data sources, e.g. filling in data gaps in MODIS AOD images from Terra with AOD observations from Aqua (Bai et al., 2019; Sogacheva et al., 2020), or simply to fuse with AOD simulation outputs from numerical models (Xiao et al., 2021). Such a substitution method is straightforward and effective, especially in an era with big earth observation data. Nonetheless, cross-mission biases among satellite-based retrievals acquired from different platforms and/or instruments are always salient due to significant differences in both instruments and retrieval algorithms. Bias correction is thus essential to reducing systematic biases (Bai et al., 2016b, 2016a), and different methods such as linear regression and maximum likelihood estimation were applied to account for cross-mission biases prior to data merging (Bai et al., 2016a, 2016b, 2019; Ma et al., 2016; Xu et al., 2015). More complex data fusion methods like the Bayesian maximum entropy (Tang et al., 2016; Wei et al., 2021b), were also applied to fuse AOD products with different spatial resolutions.

Another type of gap-filling methods work in a principle to recover missing information via dominant pattern recognition and reconstruction over space and time, and the data interpolating empirical orthogonal functions (DINEOF) method is a representative one (Beckers and Rixen, 2003; Liu and Wang, 2019). Two similar methods were developed to fill in data gaps in ground-measured $PM_{2.5}$ concentration time series and geostationary satellite-sensed AOD imageries (Bai et al., 2020; Li et



al., 2022b). Similarly, Zhang et al. (2022) developed a spatiotemporal fitting algorithm to gap-fill the daily MODIS AOD
product, with AOD values mainly predicted based on annual trend and spatial residues inferred from neighboring pixels.
Nonetheless, data gaps are hardly to be properly reconstructed simply based on a single data source, especially for those with
excessive missing values (e.g., satellite-based AOD). Retrieving the missing AOD information from diversified external data
products via various learning algorithms in artificial intelligence, e.g., numerical AOD simulations (Li et al., 2020; Xiao et al.,
2017) and even meteorological factors (Bi et al., 2019), was proven an effective and feasible way for improving spatial
coverage of reconstructed AOD fields.
Machine learning methods have been widely applied to downscale numerical AOD simulations to satellite AOD footprints,
while data gaps in satellite-based AOD imageries were then filled with downscaled data (He et al., 2023; Wei et al., 2021a).
Given the powerful approximation capacity, machine learning methods were extensively used for bias correction in gap-filling
problems over recent years (Bai et al., 2022b, 2023b; He et al., 2023; Wang et al., 2022; Wei et al., 2021a; Xiao et al., 2021).
Leveraging machine learning and tensor completion methods, i.e., a more complex big data analytics framework, was
developed to integrate six satellite-based AOD datasets and numerical aerosol diagnostics as well as in situ air quality
measurements (Bai et al., 2022a). The comparable data accuracy of reconstructed AODs well demonstrate the efficacy of this
gap-filling approach, yielding a long-term gap-free high-resolution MODIS-like AOD and PM concentration dataset (LGHAP
version 1) in China. Despite the good reconstruction performance, further investigations have recently proven that prior
information is vital for tensor-flow-based gap-filling, especially over areas with substantial missing values, and the
reconstruction results would be prone to large uncertainty with few valid observations in the input tensor (Bai et al., 2022a; Li
et al., 2022a, 2022b). Moreover, invariant background and equal weights for different AOD inputs may not only reduce the
convergence speed but degrade the reconstruction accuracy.
Leveraging an improved big earth data analytics approach, a global scale LGHAP dataset, termed as LGHAP v2 hereafter,
was hereby generated to provide daily global gap-free AOD and $PM_{2.5}$ concentrations at 1-km grid resolution as of 2000. In
order to accommodate global massive earth observations acquired from diverse satellites, numerical models, and air quality
monitoring stations, several new algorithmic improvements were applied to the tensor-flow-based gap filling approach,
including an attention-reinforced tensor construction strategy and an adaptive background information updating scheme,
aiming at improving convergence speed and mitigating modeling bias propagation in numerical AOD diagnostics. Moreover,
a novel deep learning method named as the SCene-Aware ensemble learning Graph ATtention network (SCAGAT) was
developed to fulfill global $PM_{2.5}$ concentration mapping. Benefiting from the customized algorithmic improvements and the
novel SCAGAT $PM_{2.5}$ mapping method, LGHAP v2 dataset has not only extended spatial coverage from China to global but
also improved data accuracy compared to LGHAP v1. To our knowledge, this is the first publicly accessible global long-term
gap-free MODIS-like AOD and $PM_{2.5}$ concentration dataset with daily 1-km resolution, which could be used to help deepen
our understanding of global aerosol pollution variations as well as adverse impacts on public health, ecosystem, weather, and
climate. In the following we provided a more detailed description of diversified data sources analyzed in this study as well as
versatile machine learning and deep learning methods used to manipulate big earth observational data. Performance of
algorithmic improvements as well as the data accuracy of global gap-free AOD and $PM_{2.5}$ concentration data were then
comprehensively evaluated by comparing against worldwide in-situ AOD and $PM_{2.5}$ concentration measurements.
**2 Data sources**
In the current study, we still attempt to synergistically integrate big earth data acquired from diverse sources to generate
global long-term gap-free AOD dataset with daily 1-km resolution, from which spatially contiguous $PM_{2.5}$ concentration
estimates can be derived by a more robust way to minimize the gaps and maximize the prediction accuracy. As shown in Table



1, a large variety of big earth data were hereby employed, including gridded AOD products from six polar orbiting satellites
as well as numerically simulated MERRA-2 AOD and aerosol diagnostics, eleven meteorological reanalysis fields, six datasets
of in situ AOD and air pollutants concentration measurements. Additionally, auxiliary variables representing land use and land
cover types, elevation, population density, and vegetation index were used not only to help harmonize discrepancies among
heterogeneous data prior to data integration but also to aid in global $PM_{2.5}$ concentration mapping.

### 2.1 Satellite-based AOD products

AOD retrievals derived from MODIS observations on board Terra ($AOD_{Terra}$) with the Multi-Angle Implementation of
Atmospheric Correction (MAIAC) algorithm were hereby used as the benchmark to generate global long-term gap-free AOD
dataset, given their finer spatiotemporal resolution and longer temporal coverage (Lyapustin et al., 2011, 2018; Mhawish et
al., 2019). Previous studies demonstrated a better quality of the MAIAC AOD data relative to other gridded products (Chen et
al., 2021; Martins et al., 2017; Qin et al., 2021), not only data accuracy but also spatiotemporal completeness, even better than
those retrieved with the well-known Dark Target and Deep Blue algorithms (Jiang et al., 2023; Liu et al., 2019). Figure S1
presents spatial and temporal distribution of the coverage ratio of valid $AOD_{Terra}$ from 2000 to 2021 at each satellite footprint
across the globe.
Satellite-based AOD retrievals from a few key instruments other than MODIS were applied to support gap filling of
$AOD_{Terra}$. They include: 1) Visible Infrared Imaging Radiometer Suite (VIIRS, on board Suomi-NPP), 2) Multi-angle Imaging
SpectroRadiometer (MISR, on board Terra), 3) Advanced Along-Track Scanning Radiometer (AATSR, on board Envisat), 4)
POLarization and Directionality of the Earth's Reflectance (POLDER, on board PARASOL), and 5) Sea-Viewing Wide Field-
of-View Sensor (SeaWIFS, on board SeaStar). Meanwhile, MAIAC AOD data from MODIS on board Aqua were also applied
as the complementary data set to support gap-filling of $AOD_{Terra}$. Given different overpassing times and temporal spans, these
multisensory AOD products provide complementary observations to help reduce random errors when reconstructing data gaps
in $AOD_{Terra}$ due to the increased prior knowledge. A brief summary of these AOD products can be found in Bai et al. (2022a)
and Wei et al. (2020).

### 2.2 Ground-based AOD observations and air quality measurements

### 2.2.1 AERONET AOD observations

Ground-based AOD observations from AERONET have long been used as the ground truth to validate AOD retrievals
from other instruments, especially satellite-based AOD retrievals. In this study, AOD observations from AERONET across
the globe during the study period were employed as an independent data source to validate the data accuracy of the gap-filled
AOD dataset. To guarantee adequate number of AERONET AOD samples, the Level 1.5 rather than Level 2.0 AOD
observations were applied, though the latter has stricter screening criteria for quality control. For spatial registration, each
AERONET AOD observation was spatially collocated with mean AOD values over grids within a 50 × 50 km window size.
Figure S2 presents spatial distribution of AERONET sites and air quality monitoring stations providing pivotal AOD and $PM_{2.5}$
concentration observations used in this study.

### 2.2.2 Air quality measurements

Concentrations of $PM_{2.5}$ and other relevant air pollutants like $NO_2$, $SO_2$, $PM_{10}$, CO were acquired from a few agencies
and/or monitoring centers, such as the United States Environmental Protection Agency, European Air Quality Portal, China
National Environmental Monitoring Centre, Canada National Air Pollution Surveillance, Japan National Institute for



Environmental Studies, to name a few. Moreover, air quality measurements acquired from the World's Air Pollution Index,
an open-source data hub, were included as well. $PM_{2.5}$ concentrations were used as the learning target for global $PM_{2.5}$
concentration mapping. Aiming at providing critical prior information to facilitate AOD gap-filling, ground-based air quality
measurements were also used as an important proxy for regional AOD prediction, benefitting from the relatively dense
distribution of air quality monitoring networks as well as good associations between aerosol loadings and regional air pollutants
concentrations.

Atmospheric visibility, a common air quality indicator that is highly associated with aerosol loadings, were acquired from
worldwide meteorological monitoring stations and used as the critical predictor like air pollutants concentrations to predict
AOD over each monitoring site via data-driven modeling. Given much denser distribution of ambient air quality and
meteorological monitoring sites, as shown in Figure S2 for the spatial distribution of global air quality and meteorological
monitoring sites used in this study, as well as the good accuracy of site-based AOD predictions (Bai et al., 2022b; Li et al.,
2022b), a global virtual AOD monitoring network was established, providing us with an unparallel opportunity to improve
AOD gap-filling accuracy, especially for regions being disturbed by massive satellite AOD data voids.
**2.3 Numerical simulations**
**2.3.1 MERRA-2 aerosol diagnostics**

Despite the coarse spatial resolution and large modeling bias, the Modern-Era Retrospective Analysis for Research and
Applications, version 2 (MERRA-2) aerosol diagnostics including AOD and chemical components like black carbon, organic
carbon, dust, and sulfate aerosols were employed to provide prior information to advance AOD gap-filling. As the NASA's
latest reanalysis for the satellite era, MERRA-2 is generated using the newly Earth system model of Goddard Earth Observing
System, version 5 (GEOS-5), providing global simulations of a variety of geophysical and chemical variables on the Earth
surface. More detailed descriptions of the assimilation system and the data quality of MERRA-2 aerosol reanalysis can be
found in the literature such as Buchard et al. (2017) and Randles et al. (2017). By taking $AOD_{Terra}$ into account as a learning
target, data-driven models were established to downscale MERRA-2 AOD to the level of $AOD_{Terra}$, with MERRA-2 aerosol
diagnostics as well as meteorological, geographical, and socioeconomic factors used as covariates. The downscaling model
not only improves the spatial resolution but also corrects large modeling biases in MERRA-2 AOD. Given the global complete
coverage merit, the downscaled gap-free AOD data were then used as critical prior information to facilitate AOD gap-filling,
in particular over regions lacking observational AOD.
**2.3.2 ERA-5 reanalysis**

As the latest atmospheric reanalysis produced by the European Center for Medium Weather Forecast, ERA-5 provides
hourly estimates of a variety of atmospheric, terrestrial, oceanic, climatic and meteorological variables. The data are provided
at about 30 km grid resolution on the Earth surface resolving the atmosphere using 137 levels from the surface up to a height
of 80 km, covering the period from January 1940 to the present (Hersbach et al., 2020). Atmospheric parameters including
surface pressure, air temperature, relative humidity, wind speed, total column water, total precipitation, surface solar radiation
downward, instantaneous moisture flux, and boundary layer height were retrieved from ERA-5 and used as important modeling
covariates, not only in data harmonization models to calibrate other AOD and relevant data products to the level of $AOD_{Terra}$,
but also in global $PM_{2.5}$ mapping models to help approximate nonlinear associations between $PM_{2.5}$ and AOD. Bilinear
interpolation was applied to map ERA-5 reanalysis data down to the $AOD_{Terra}$ footprint for spatial registration.





**Table 1.** Summary of diverse big earth data used in this study to help generate global gap-free AOD dataset at daily/1-km
resolution (LGHAP v2) from 2000 to 2021.

| Category | Dataset | Temporal resolution | Spatial resolution | Time period |
|---|---|---|---|---|
| AOD | MCD19A2 | daily | 1-km | 2000–2021 |
| | Terra/MISR | daily | 4.4-km | 2000–2021 |
| | NPP/VIIRS | daily | 5-km | 2012–2021 |
| | Envisat/AATSR | daily | 10-km | 2000–2012 |
| | PARASOL/POLDER | daily | 10-km | 2005–2013 |
| | SeaWiFS/OrbView-2 | daily | 10-km | 2000–2010 |
| | AERONET | hourly | / | 2000–2021 |
| Meteorological factors | Air temperature | hourly | | |
| | U/V component of wind | hourly | | |
| | Relative humidity | hourly | | |
| | Surface pressure | hourly | | |
| | Boundary layer height | hourly | 0.25° | 2000–2021 |
| | Total column water vapor | hourly | | |
| | Surface solar radiation downwards | hourly | | |
| | Total precipitation | Hourly | | |
| | Instantaneous moisture flux | hourly | | |
| | Visibility | 3-hour | / | 2000–2021 |
| Air quality measurements | $PM_{2.5}$, $PM_{10}$, $NO_2$, $SO_2$, CO | hourly | / | 2000–2021 |
| Population | WorldPop | annual | 1-km | 2000–2020 |
| Land cover | Impervious (GISA) | annual | 30-m | 2000–2020 |
| | MCD12Q1 | annual | 500-m | 2000–2021 |
| NDVI | MOD13A3 | monthly | 1-km | 2000–2021 |
| Aerosol diagnostics | MERRA-2 | hourly | 0.5°×0.625° | 2000–2021 |
| Elevation | SRTM DEM | / | 90 m | / |

**2.4 Auxiliary data**
Several socioeconomic and geographic factors were also applied as covariates to support predictions of AOD and $PM_{2.5}$
concentration. Gridded population data from WorldPop were used to indicate spatial distribution of residents, which were
applied as a proxy of anthropogenic aerosol emission intensity. To resolve land use dependent aerosol emissions, land cover
types and vegetation index derived from MODIS observations as well as the coverage ratio of impervious surface at the
$AOD_{Terra}$ footprint were also applied. Digital elevation data collected from the Shuttle Radar Topography Mission (SRTM)
with a resolution of 1 arc-second were used to characterize potential impacts of topography on aerosol loadings.
**3 Methods**
**3.1 Tensor-flow-based AOD reconstruction**
**3.1.1 Overview of AOD gap-filling method**
Deriving spatially contiguous $PM_{2.5}$ concentrations from gap-filled AOD images has been proven more promising for a
better spatial analysis of large-scale $PM_{2.5}$ distribution (Bai et al., 2022b). In this study, the big earth data analytics proposed
in Bai et al. (2022a) was further adapted for generating global gap-free AOD imageries to support various content-based



mapping. Figure 1 presents the workflow of the improved framework of the big earth data analytics for generating global gap-
filled MODIS-like AOD maps. This framework consists of three primary data manipulation procedures including: 1) machine
learned multimodal data homogenization, 2) knowledge-reinforced AOD tensor compiling, and 3) tensor-flow-based AOD
reconstruction. This improved big earth data analytics approach empowered us to weave multimodal AODs and versatile big
earth observations from diversified sources together neatly via a synergy of state-of-the-art machine learning and tensor
completion methods. Since the technical flow of this big earth data analytics framework was well elaborated in Bai et al.
(2022b), we only provided an overview of this method while emphasizing the newly developed algorithmic components in the
following.

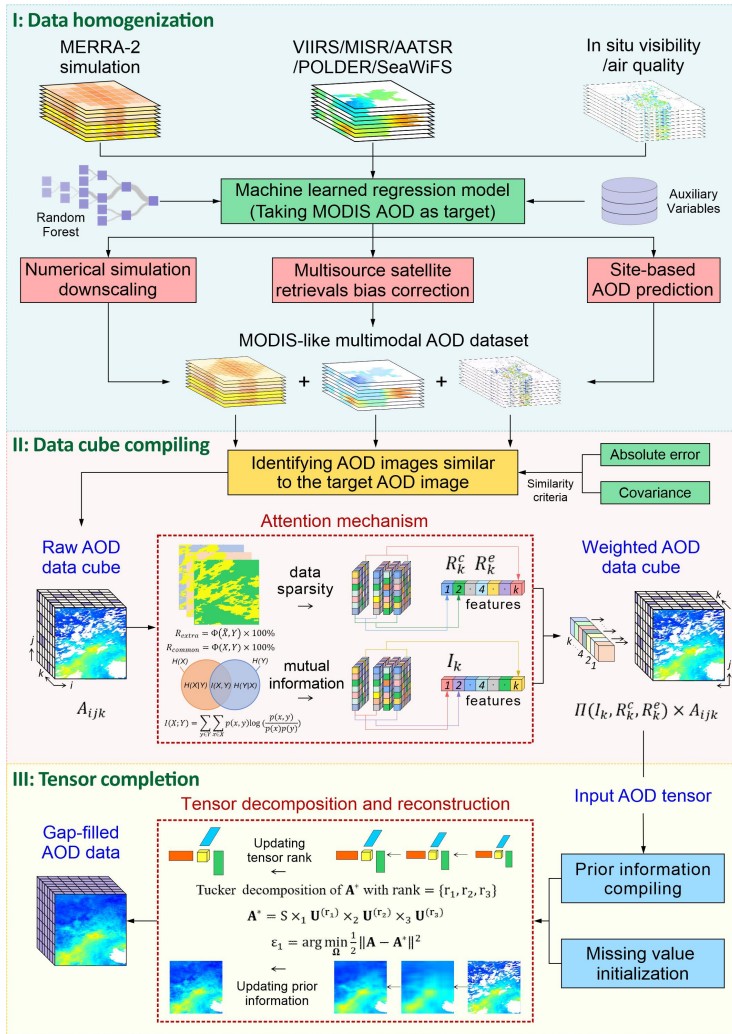


**Figure 1.** A schematic illustration of the enhanced big earth data analytics for generating MODIS-like global gap-free AOD dataset.
Leveraging random forest-based regression models, multimodal AODs and relevant aerosol data acquired from different
satellites, ground monitoring stations, and numerical models were firstly harmonized to resemble the baseline dataset of
AOD$_{Terra}$, aiming at not only minimizing cross-sensor biases arising from algorithmic differences but also accounting for spatial
heterogeneities due to different spatial resolutions. This data homogenization process is vital for the tensor-flow-based AOD



gap-filling because the bias-corrected and downscaled AOD estimates were critical inputs to form AOD data cube. More
details related to multisource data homogenization were described in Text S1 in the supporting information. AOD data cube
was then created based on homogenized data at each individual data tile. A proper AOD data cube compiling is undoubtedly
essential for the tensor-flow-based AOD reconstruction. To fill data gaps in each individual $AOD_{Terra}$ image, an AOD data
cube was constructed, in our previous gap-filling framework, by simply aggregating harmonized multisensory AOD data on
the same date along with historical $AOD_{Terra}$ images resembling similar spatial patterns over the same region. Due to excessive
nonrandom missing values in $AOD_{Terra}$ imageries, both downscaled MERRA-2 AOD grids and AOD estimates derived from
air quality and visibility measurements were used conjunctively to identify historical $AOD_{Terra}$ imageries with a similar spatial
distribution. The selected historical $AOD_{Terra}$ images and bias-corrected AOD images from other satellites on the same date
were individually incorporated as a slice of the tensor. Additionally, dispersed in situ AOD estimates and 5% randomly selected
AOD estimates from the downscaled MERRA-2 data were directly overlaid onto the corresponding $AOD_{Terra}$ grids where valid
retrievals were not present. These implementations not only helped improve the gap-filling accuracy but also boosted the
convergence speed given the provision of prior knowledge.
High order singular value decomposition (HOSVD), an orthogonal Tucker decomposition method, was finally applied to
each compiled AOD data cube for tensor-flow-based pattern recognition and tensor completion. Data gaps within the input
AOD tensor were firstly filled with the spatial average of each individual AOD image to initiate tensor decomposition. The
AOD tensor was then decomposed along every two-dimension of AOD tensor independently, and a new tensor was
subsequently reconstructed based on the principal modes learned along every two-dimension of the tensor via a low-rank
approximation (i.e., generating an approximating matrix with reduced rank for compression). During the tensor reconstruction
process, $AOD_{Terra}$ observations in the target image to be gap-filled were deemed as the hard data (i.e., true state and invariant
throughout the tensor completion procedure) while multisensory AOD estimates and historical $AOD_{Terra}$ images were used as
the soft data (prior information and updated by iterates till convergence). By iteratively adjusting dimension-varied ranks, data
values over grids to be gap-filled were updated and tuned to optimize both spatial homogeneity and information entropy
concurrently (Bai et al., 2020, 2022a). This tensor completion process continued till reaching a good agreement (with a bias
decay ratio <0.1%) between reconstructed values and priorly reserved $AOD_{Terra}$ observations.

### 253   3.1.2 Algorithmic improvements

To accommodate massive data analytics for global-scale AOD gap-filling, two major algorithmic enhancement modules
were incorporated to help improve the reconstruction efficiency and accuracy, focusing on optimizing data manipulation
procedures in tensor-flow-based AOD gap filling. Rather than treating each slice of data in raw AOD data cube equally, an
attention mechanism was introduced to optimize AOD tensor compiling, aiming at underscoring the importance of those AOD
imageries with fewer data gaps while more closely resembling the target $AOD_{Terra}$ imagery during tensor-flow-based AOD
reconstruction. Meanwhile, an adaptive prior information updating scheme was implemented to help mitigate the propagation
of large modeling biases in numerical AOD diagnostics to the final reconstructed fields during the tensor reconstruction
procedure. Moreover, the rank updating strategy was optimized to improve computing efficiency in tensor completion. The
algorithm 1 below presents the pseudo code of the optimized algorithm used for tensor-flow-based AOD reconstruction.

### 263   3.1.2.1 Attention-reinforced AOD tensor construction

Both the target data (i.e., $AOD_{Terra}$ image to be gap-filled) as well as soft data (i.e., AOD estimates from other data sources
and historical $AOD_{Terra}$ imageries) in AOD tensor were treated equally during the tensor decomposition and reconstruction
process in our previous tensor completion framework as shown in Bai et al. (2022a). Such an indifferent data treatment not



only neglected the information abundance of soft data but also ignored the similarity of spatial patterns between soft and target
data, leading the reconstructed field more likely to resemble the dominant patterns learned from imageries with fewer gaps,
rather than images with higher similarities to the target data. To account for this drawback, an attention mechanism was
implemented to weigh each slice of data in the input AOD tensor, aiming at improving the AOD reconstruction performance
by learning from spatiotemporal features embedded in more relevant data fields rather than all available data.

As a widely used technique in deep learning regimes, attention mechanism is a mimic of cognitive attention allowing the

model to focus on specific parts of the input data, achieved by assigning higher weights to more crucial elements in ensemble
learning. Regarding the tensor-flow-based AOD reconstruction task, data slices with higher similarity to the target image and
fewer data gaps should play more important roles than those less similar ones with extensive data gaps in tensor completion.
Three statistical metrics, i.e., mutual information (Shannon, 1948), spatial coverage ratio of common observations ($R_{common}$)
between each soft data and hard data, and spatial coverage ratio of extra observations beyond common observations in soft
data ($R_{extra}$), were calculated to determine the weight assigned to each data slice of the input AOD tensor. Below gives the
formulas to calculate these three statistical metrics.
$$MI(X,Y) = \sum_{y\in Y}\sum_{x\in X} p(x,y)\log\left(\frac{p(x,y)}{p(x)p(y)}\right) \tag{3}$$

$$R_{common} = \Phi(X,Y) \times 100\% \tag{4}$$

$$R_{extra} = \Phi(\tilde{X},Y) \times 100\% \tag{5}$$

where $X$ and $Y$ refer to common observations in soft and hard data, respectively. $\tilde{X}$ denotes extra observations in soft data.
$p(x,y)$ is the joint probability mass function of $X$ and $Y$, $p(x)$ and $p(y)$ are the marginal distribution mass function of $X$ and
$Y$, respectively. $\Phi(X,Y)$ is the spatial coverage ratio of the common observations, and $\Phi(\tilde{X},Y)$ is the spatial coverage ratio
of extra observations in the soft data. By multiplying these three normalized weights to the corresponding soft data, an
attention-reinforced AOD tensor was constructed in turn, which was then used as the input data cube for tensor completion.
**Algorithm 1**. The pseudo code of the optimized algorithm used for tensor-flow-based AOD reconstruction.

---

**Input:** tensor $\mathbf{A} \in \mathbf{R}^{N_1 \times N_2 \times N_3}$ with $\boldsymbol{\Omega} = \{(i,j,k): A_{ijk} \text{ is observed}\}$, threshold $T_1, T_2$

**Output:** reconstructed entries $\mathbf{A}' = \mathbf{A}^*(:,:,k^t) \in \mathbf{R}^{N_1 \times N_2}$

1: Attention mechanism: $\omega_k = \Pi(MI_k, R_k^c, R_k^e)$

2: Initialize $A_{ijk}^* = \begin{cases} \omega_k \cdot A_{ijk} & (i,j,k) \in \boldsymbol{\Omega} \\ \sum_i \sum_j A_{ijk} & (i,j,k) \notin \boldsymbol{\Omega} \end{cases}$

3: **for** $r_3 = \frac{1}{3}N_3$ to 1 step -2 **do**

4:      $n_1 = n_2 = 0$

5:      **while** $\varepsilon_1 > T_1$ or $(n_1 < \frac{1}{3}N_1$ and $n_2 < \frac{1}{3}N_2)$ **do**

6:          $n_1 = n_1 + 1, n_2 = n_2 + 1$

7:          $r_1 = \frac{n_1 N_1}{75}, r_2 = \frac{n_2 N_2}{75}$

8:          $\mathbf{A}^* = HOSVD(\mathbf{A}^*, rank = \{r_1, r_2, r_3\})$:

9:          $\mathbf{A}^* = S \times_1 \mathbf{U}^{(r_1)} \times_2 \mathbf{U}^{(r_2)} \times_3 \mathbf{U}^{(r_3)}$

10:         $\varepsilon_1 = \arg\min_{\boldsymbol{\Omega}} \frac{1}{2}\|\mathbf{A} - \mathbf{A}^*\|^2$

11:         $\mathbf{A}_{\boldsymbol{\Omega}}^* = \mathbf{A}_{\boldsymbol{\Omega}}$

12:         $\mathbf{A}_{\tilde{\boldsymbol{\Omega}}}^* = \omega_1 \mathbf{A}_{\tilde{\boldsymbol{\Omega}}}^* + \omega_2 \mathbf{A}_{\tilde{\boldsymbol{\Omega}}}$, $\tilde{\boldsymbol{\Omega}}$ denotes background location

13:     **end while**

14:     **if** $\arg\min_{\boldsymbol{\Omega}} \frac{1}{2}\|\mathbf{A} - \mathbf{A}^*\|^2 < T_2$ **then**

15:         break;

16:     **end if**

17: **end for**

---



### 3.1.2.2 Adaptive prior information updating

To facilitate AOD gap-filling over regions with abundant data gaps, in our previous method, 5% random samples from the downscaled MERRA-2 AOD image ($AOD_{M2}$ hereafter) on the same date were used as prior information and placed directly onto grids without observational AOD (i.e., $AOD_{Terra}$ and site-based AOD estimates from air quality and visibility measurements). Although this empowered us to improve the convergence speed during tensor completion, spatial patterns of the reconstructed field over regions with excessive data gaps were more likely to resemble the distribution of $AOD_{M2}$ given an equal weight of the soft and hard data. In other words, sparse observational AODs derived from air quality measurements played a relatively weak role in tensor completion when confronting with $AOD_{M2}$. In such a context, large modeling biases in $AOD_{M2}$ might be introduced into the final reconstructed fields.

In this study, we introduced an adaptive prior information updating scheme to help mitigate potential bias propagation from $AOD_{M2}$. Differing from the strategy used in our previous method, the AOD prior information in the input AOD tensor was also forced to update by iterations, rather than maintaining them invariant as $AOD_{Terra}$ observations throughout the tensor completion process. Specifically, random $AOD_{M2}$ samples were only used to initiate the tensor construction, while weighted averages of these prior information and the corresponding reconstructed values were then used as new prior information for the next iteration. Meanwhile, weights assigned to the reconstructed fields were gradually increased by iteration till convergence. The ultimate goal was to improve the contribution of reconstructed fields learning from actual observations while reducing the influence of $AOD_{M2}$. The ablation experiments also demonstrated that such a scheme is effective in mitigating bias propagation from $AOD_{M2}$, largely improving the reconstruction performance over regions with limited observational data.

### 3.1.2.3 Optimized global data tile partition and rank updating

Given high spatial and temporal resolution of $AOD_{Terra}$ imageries, performing global-scale AOD gap-filling is thus challenging due to huge computation burdens. To improve the computational efficiency and to make the computing workload manageable, the following algorithmic improvements were applied. Firstly, global $AOD_{Terra}$ data over land were divided into 480 data tiles, with AOD gap-filling performed over each data tile independently. The size of a tile was determined empirically after performing a set of gap-filling trials with different sizes, and a nominal size of a tile covering 700×700 pixels (could be different over coastal regions) was finally applied to balance the computing workload and the learning accuracy. Figure S3 presents spatial distribution of optimized data tiles used in this study for global AOD gap-filling. Moreover, a 50-pixel overlap on the boundary of each tile was enforced, and an inverse distance weighting scheme was finally applied to these overlapped pixels when mosaicking the gap-filled tiles to eliminate the boundary effect between tiles toward a smooth distribution of AOD across the globe.

An optimized rank updating strategy was also proposed to improve the learning efficiency. In tensor completion process, tensor's decomposition and reconstruction are driven by iteratively updating tensor ranks. To improve the computational efficiency of global AOD gap-filling, we developed an optimized strategy to update ranks between iterations. Specifically, the ranks were updated in an ascending order along with the first and second dimensions in the inner loops to enhance spatial details of reconstructed AOD. In contrast, ranks were updated in a descending fashion along with the third dimension in the outer loop to aggregate the target $AOD_{Terra}$ image with soft data in a low-rank approximation manner.

### 3.2 Global PM₂.₅ concentration modeling

The sparse and uneven distribution of ground-based air quality monitoring stations poses significant challenges to global $PM_{2.5}$ concentration mapping, especially over regions of fewer $PM_{2.5}$ concentration measurements (e.g., Africa and south



America in Figure S2). Also, how to reinforce the spatial representativeness of data-driven models when extrapolating them
over space is elusive. As a novel idea, SCAGAT was developed and applied to better estimate global $PM_{2.5}$ concentration from
gap-filled AOD imageries by accounting for spatial representativeness of each data-driven model. Rather than establishing a
global $PM_{2.5}$ estimation model using all available data pairs collected from worldwide monitoring stations, site-specific $PM_{2.5}$
estimation models were firstly developed using random forest over each air quality monitoring station with long-term $PM_{2.5}$
concentration measurements. For a given grid, raw $PM_{2.5}$ concentration estimates were then estimated from a set of independent
site-specific $PM_{2.5}$ estimation models, of which should resemble similar geographic scene features as the given grid cell, under
the assumption that the relationship between AOD and $PM_{2.5}$ is similar over regions with analogue environmental background.
Nine distinct factors covering geodetic location, land cover types, climate zones, AOD levels, and population density were
utilized to characterize scene attributes of each grid cell. Subsequently, a graph attention network was used to aggregate these
raw $PM_{2.5}$ estimates to better predict $PM_{2.5}$ concentration over the target grid cell, with weights assigned to the adjacency
matrix in reference to the differences between nine different scene features while the node bias was given as the testing
accuracy of each site-specific $PM_{2.5}$ prediction model. Figure S4 presents the workflow of the proposed SCAGAT model for
global $PM_{2.5}$ concentration mapping. This novel ensemble learning method enables us to better predict $PM_{2.5}$ concentrations
across the globe, especially over regions with few or even none in situ $PM_{2.5}$ concentration measurements. More details of the
SCAGAT model were introduced in Text S2 as part of the supplementary information.

## 4  Results

### 4.1 Efficacy assessment of algorithmic enhancement modules

Ablation experiments were firstly conducted to evaluate the accuracy improvement potential of each newly developed
algorithmic enhancement module. Three case studies were simulated by masking actual $AOD_{Terra}$ retrievals with randomly
selected cloud masks on different dates, and methods reinforced with different enhancement modules were then applied to
reconstruct priorly held-out AOD values. For inter-comparison, the AOD gap-filling framework developed by Bai et al.
(2022a), was thus used for benchmarking. As shown in Figure 2, AOD distributions reconstructed with methods embedding
attention mechanism and/or adaptive background information updating modules better resembled actual $AOD_{Terra}$ retrievals
than the benchmark method, justifying the efficacy of these two enhancement modules. Given an equal weight of each slice
of data in the input AOD tensor, the reconstructed data fields from the benchmark method were prone to resembling a mean
state determined largely by the principal mode of the input tensor. In this context, peak and/or low values in the target image
might be underestimated (or overestimated for low values) if with relatively few soft data resembling similar patterns in the
input tensor (refer to the third panel in Figure 2).
With the involvement of the attention mechanism, each slice of data in raw AOD data cube was weighted adaptively,
with larger weights given to data slices not only having larger spatial coverage but also with higher similarities to the target
$AOD_{Terra}$ image. This strategy is vital to reducing contributions from irrelevant data, especially when facing with unbalanced
data samples in raw AOD data cube, i.e., more irrelevant data and fewer similar imageries. Moreover, the importance of the
target image was maximized during the tensor completion procedure by giving a 100% weight. Compared to the benchmark
method, peak and/or low values in raw $AOD_{Terra}$ images were better reconstructed by the method embedding the attention
mechanism. For instance, low AOD values in the north in Figure 2b were apparently overestimated by the benchmark method,
whereas such effect was largely mitigated using methods involving the attention mechanism.
In contrast to the benchmark by using an invariant background throughout the tensor completion, an adaptive background
updating scheme was thus applied to not only accelerate the convergence speed but also mitigate possible error propagation



from numerical simulations to the final reconstructed fields. As illustrated in Figure S5, compared to the benchmark, the
manually added outliers in raw background fields were better detected and reconciled by the improved method owing to the
involvement of adaptive background updating module, avoiding large error propagation from background fields into the
reconstructed AOD data. The better quality of reconstructed fields derived from improved methods well demonstrate the
efficacy of two newly developed algorithmic enhancement modules. Nevertheless, as compared in Figure 2c, the benefits of
these two enhancement modules were largely cancelled when dealing with images with excessive data gaps, showing a
marginal accuracy improvement relative to the benchmark method. The inherent reason could be attributable to few
observational data in the target image for reference to leverage attention mechanism.

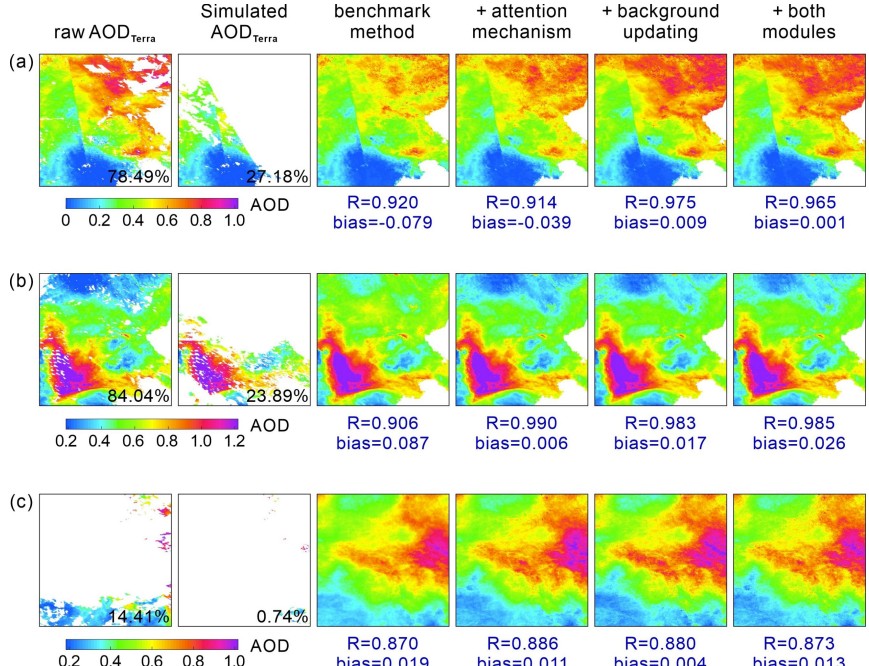

**Figure 2.** Performance evaluation of different algorithmic enhancement modules on the reconstructed AOD distribution. Raw $AOD_{Terra}$ denotes actual AOD retrievals from Terra, while simulated $AOD_{Terra}$ refers to partially masked $AOD_{Terra}$. The benchmark method is the AOD gap-filling approach proposed in Bai et al. (2022a). The latter three columns present the reconstructed fields using the enhanced benchmark method. R and bias denote correlation coefficient and deviations between observed and reconstructed AOD data, respectively. Percent numbers shown in the two left panels indicate spatial coverage ratio of valid AOD retrievals over the selected scenes.

In Figure 3 we evaluated impacts of missing rate on the AOD gap-filling accuracy. By masking raw $AOD_{Terra}$ retrievals
with arbitrarily selected cloud masks, $AOD_{Terra}$ images under different missing rates were generated and used as target images
for gap-filling (i.e., images in the top panel). The results show good agreements between observed and reconstructed AOD
fields, even over extreme situations with excessive data gaps, demonstrating an excellent performance of the proposed gap-
filling method. As expected, the reconstruction accuracy decreased along with an increase in missing rate. For instance, the
low values in the upper left in raw $AOD_{Terra}$ image were not properly reconstructed when missing rate was greater than 80%,
highlighting the vital importance of prior information on the gap-filling accuracy. Therefore, increasing prior information is
the most promising way to improve the accuracy in gap-filling, in particular for those areas with substantial data gaps.

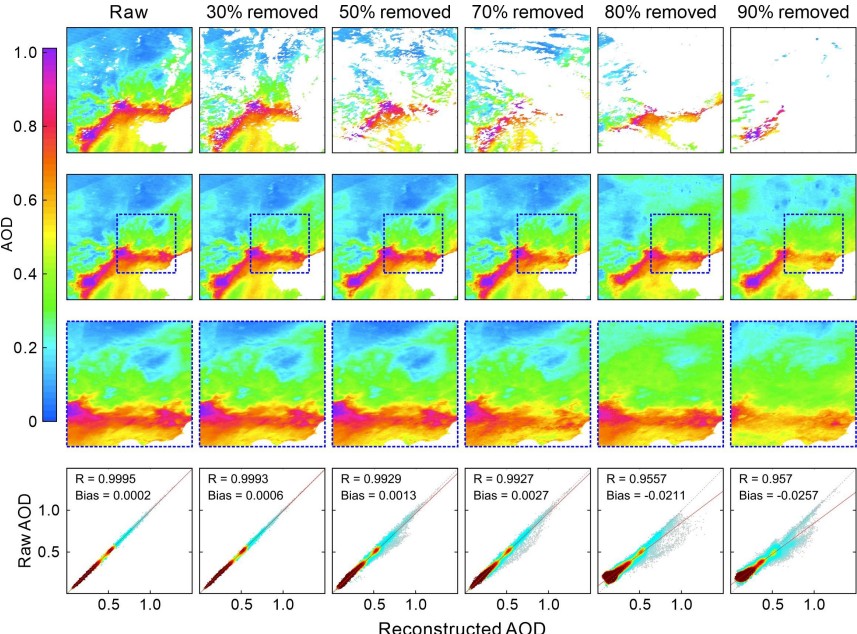

**Figure 3.** Impacts of missing rate on the AOD gap-filling accuracy. Numbers on the top indicate the percentage of removed AOD data in raw AOD$_{Terra}$ image (top panel). The second row shows the distribution of gap-filled AOD with zoom in maps present in the third row. The bottom panel presents scatter plots between observed AOD (raw data) and AOD reconstructed from different inputs.

### 4.2 Data accuracy of global gap-free AOD in LGHAP v2

By comparing against independent AOD observations from AERONET, the data accuracy of gap-free AOD in LGHAP v2 was comprehensively evaluated across the globe. Figures. 4a–c present spatial distribution of site-specific correlation coefficient (R), root mean square error (RMSE), and bias between reconstructed AOD and AOD observations from AERONET, respectively. Regardless of the uneven distribution of ground monitoring stations and the difference in data samples between sites, the ground validation results indicate good agreements between AOD in LGHAP v2 and AERONET observations, with an average of site-specific correlation coefficient of 0.76 and RMSE of 0.09 at the global scale. Meanwhile, the results indicate that site-specific data accuracy metrics exhibit notable spatial heterogeneities across the globe, with larger bias mainly observed in central and east Asia as well as Africa where often suffer from high aerosol loadings.

Figures. 4d–4i present scatter plots between gap-free AOD and AERONET observations at six major continental regions. The distinct accuracy metrics across regions also indicate significant spatial heterogeneities in AOD data accuracy. When compared against AOD observations from AERONET, reconstructed AOD estimates were prone to underestimate large AOD observations (>0.80) whereas overestimate low values (<0.2) across these six regions. Such an effect is particularly common in machine learning, largely due to the imbalanced distribution of data values in training samples (Johnson & Khoshgoftaar, 2019; Shi et al., 2022). Likewise, the inherent reasons for this effect in tensor completion might be identical, which could be largely attributable to the principle of low-rank approximation to fulfil tensor reconstruction and imbalanced (i.e., few extremes) AOD values in the input tensor. Consequently, the missed AOD extremes were hardly to be reconstructed to their nominal levels. Rather, the reconstructed values were inclined to resemble a mean state that was determined by principal modes due to the imbalanced data distribution.

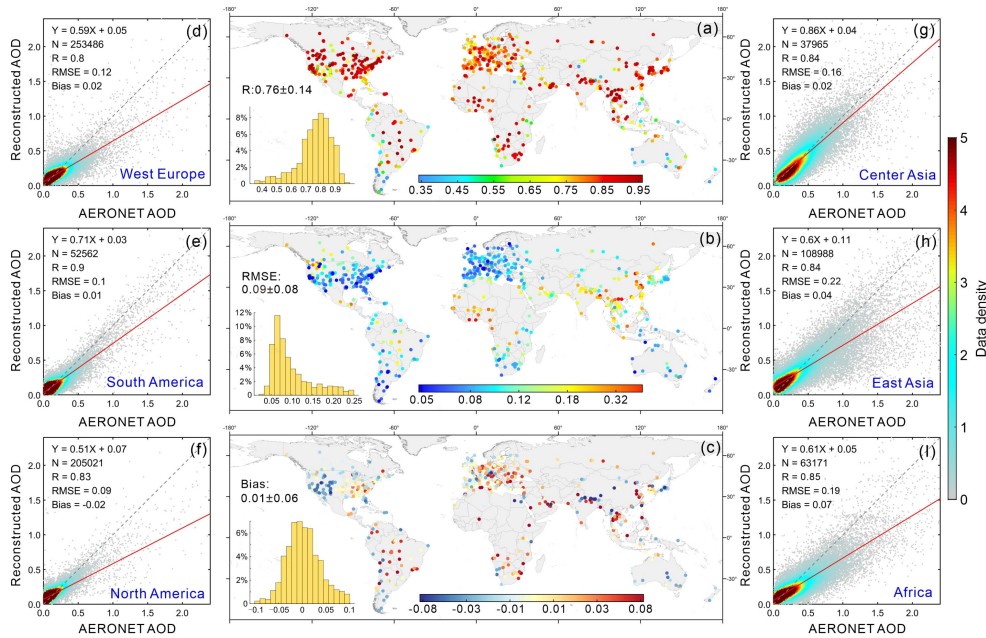

**Figure 4.** Data accuracy of daily gap-free AOD grids in LGHAP v2 dataset by comparing against AOD observations from AERONET across
the globe during 2000–2021. Note AERONET AOD observations were independent data from the gap-filling process.

To verify the data accuracy of imputed AOD estimates, we further compared the data accuracy of gap-filled AODs in
LGHAP v2 dataset with two major gridded products, i.e., satellite-based AOD retrievals from Terra (MCD19A2) and
downscaled MERRA-2 AOD (AOD$_{M2}$). As shown in Table 2, the purely reconstructed AOD estimates have a R of 0.83 and
RMSE of 0.15 compared against AERONET AOD observations at the global scale, comparable to the data accuracy of AOD$_{M2}$
(R=0.83, RMSE=0.14) but lower than that of AOD$_{Terra}$ (R=0.88, RMSE=0.11). Nevertheless, the imputed AOD estimates
achieved comparable data accuracies as AOD$_{Terra}$ in Africa (R=0.80, RMSE=0.20) and Australia (R=0.62, RMSE=0.08),
largely due to abundant satellite-based AOD retrievals over these two areas (refer to AOD coverage ratio shown in Figure S1)
to facilitate AOD gap-filling via tensor completion. In contrast, the imputed AOD estimates in Europe and Asia have poorer
data accuracies with relative to AOD$_{Terra}$, especially in Asia. Possible reasons could be ascribed to not only extensive missing
values but also significant spatial variations in aerosol loadings as well as severe aerosol pollution levels over these regions.

The gap-free AOD dataset (LGHAP v2) was generated by filling in data gaps in satellite-based AOD retrievals
(MCD19A2) with reconstructed AOD estimates at each collocated footprint over land. Ground validation results indicate that
the gap-filled AOD data in LGHAP v2 are in a good agreement with AERONET AOD observations, with R of 0.85 and RMSE
of 0.14 across the globe (Table 2), slightly worse than that of raw MCD19A2 (R=0.88 and RMSE=0.11) but higher than that
of AOD$_{M2}$ (R=0.83 and RMSE=0.14). This data accuracy outperforms that of the gap-filled AOD dataset (R²=0.6031 and
RMSE=0.1350) generated by Guo et al. (2023), in which missing AODs in MCD19A2 were predicted with versatile proxy
variables (e.g., meteorological factors and population density) via random forest. Moreover, compared to raw MCD19A2
retrievals, gap-filled AOD data in LGHAP v2 tended to overestimate AERONET AOD observations (17.59% versus 11.45%
above the envelope of expected error), implying a greater number of large AOD values were reconstructed in imputed AOD
estimates. This could be also evidenced by larger global mean AOD values (0.19) in LGHAP v2 dataset than that of MCD19A2
(0.17).





In Figure 5 we compared temporal variations in AOD between LGHAP v2 and AERONET observations at six aerosol
observing sites with long-term monitoring records. Compared to discrete AOD observations from AERONET, gap-free AOD
time series well reconstructed long-term variations of aerosol loading from 2000 to 2021 at these six monitoring sites, with R
ranging 0.83−0.97 and RMSE varying between 0.04 and 0.24. Larger RMSEs at Alta Floresta and Beijing sites are more likely
ascribed to the reconstruction failures of extreme AOD peaks. Referring to histograms of AOD deviations between LGHAP
v2 and AERONET, more than 80% of AOD biases were found to vary between −0.1 and 0.1, demonstrating a high accuracy
of gap-free AOD in LGHAP v2.
**Table 2.** Inter-comparison of AOD data accuracy between satellite-based retrievals (MCD19A2), numerical aerosol diagnostics
(MERRA-2), reconstructed data, and the final gap-free product by comparing against AOD observations from AERONET
across the globe during 2000−2021. Note reconstructed data refer to imputed AOD estimates while LGHAP v2 refers to the
gap-filled AOD dataset combining both satellite-based retrievals and reconstructed data. The expected error (EE) envelope for
AOD over land was defined as $1.5 \times AOD_{AERONET} \pm 0.05$.

| AOD Dataset | Region | Mean AOD | Number of monitors | Number of samples | R | RMSE | Bias | Below EE (%) | Within EE (%) | Above EE (%) |
|---|---|---|---|---|---|---|---|---|---|---|
| MCD19A2 ($AOD_{Terra}$) | Global | 0.17 | 1335 | 402886 | 0.88 | 0.11 | 0.02 | 13.95 | 74.59 | 11.45 |
| | North America | 0.11 | 433 | 112438 | 0.83 | 0.08 | -0.01 | 4.62 | 80.93 | 14.44 |
| | South America | 0.11 | 81 | 28265 | 0.94 | 0.07 | 0.02 | 14.17 | 75.85 | 9.97 |
| | Europe | 0.11 | 208 | 96715 | 0.80 | 0.06 | 0.02 | 11.29 | 82.22 | 6.49 |
| | Asia | 0.31 | 321 | 90821 | 0.90 | 0.14 | 0.02 | 18.79 | 68.22 | 12.99 |
| | Africa | 0.21 | 110 | 48877 | 0.81 | 0.19 | 0.06 | 31.45 | 57.11 | 11.44 |
| | Australia | 0.09 | 28 | 12427 | 0.62 | 0.07 | -0.01 | 6.16 | 75.34 | 18.49 |
| Downscaled MERRA-2 ($AOD_{M2}$) | Global | 0.18 | 1335 | 811438 | 0.83 | 0.14 | 0.02 | 11.76 | 78.98 | 9.26 |
| | North America | 0.12 | 433 | 216264 | 0.80 | 0.09 | 0.00 | 5.71 | 86.22 | 8.07 |
| | South America | 0.13 | 81 | 49721 | 0.90 | 0.11 | 0.02 | 12.87 | 81.64 | 5.49 |
| | Europe | 0.13 | 208 | 177125 | 0.79 | 0.07 | 0.01 | 8.54 | 86.07 | 5.39 |
| | Asia | 0.29 | 321 | 175781 | 0.78 | 0.24 | 0.06 | 22.54 | 65.14 | 12.32 |
| | Africa | 0.24 | 110 | 88374 | 0.85 | 0.15 | 0.02 | 16.13 | 67.59 | 16.28 |
| | Australia | 0.10 | 28 | 21051 | 0.76 | 0.06 | -0.02 | 2.44 | 83.60 | 13.96 |
| Reconstructed $AOD_{Terra}$ | Global | 0.21 | 1335 | 449452 | 0.83 | 0.15 | 0.01 | 12.21 | 65.52 | 22.27 |
| | North America | 0.16 | 433 | 129716 | 0.80 | 0.10 | -0.02 | 5.23 | 67.52 | 27.25 |
| | South America | 0.17 | 81 | 30073 | 0.88 | 0.11 | 0.00 | 10.51 | 67.11 | 22.38 |
| | Europe | 0.16 | 208 | 107961 | 0.73 | 0.09 | 0.00 | 9.63 | 73.63 | 16.74 |
| | Asia | 0.33 | 321 | 107876 | 0.81 | 0.24 | 0.03 | 18.64 | 56.60 | 24.76 |
| | Africa | 0.27 | 110 | 31568 | 0.80 | 0.20 | 0.06 | 29.57 | 53.88 | 16.55 |
| | Australia | 0.13 | 28 | 9628 | 0.62 | 0.08 | -0.03 | 4.60 | 64.62 | 30.77 |
| LGHAP v2 | Global | 0.19 | 1335 | 756166 | 0.85 | 0.14 | 0.01 | 12.96 | 69.44 | 17.59 |
| | North America | 0.13 | 433 | 216055 | 0.82 | 0.09 | -0.01 | 4.86 | 73.12 | 22.02 |
| | South America | 0.14 | 81 | 49707 | 0.90 | 0.10 | 0.01 | 12.57 | 71.08 | 16.34 |
| | Europe | 0.13 | 208 | 176959 | 0.76 | 0.08 | 0.01 | 10.24 | 77.40 | 12.36 |
| | Asia | 0.32 | 321 | 175728 | 0.83 | 0.21 | 0.03 | 19.08 | 61.40 | 19.52 |
| | Africa | 0.23 | 110 | 75110 | 0.81 | 0.19 | 0.06 | 29.61 | 56.64 | 13.75 |
| | Australia | 0.11 | 28 | 21048 | 0.63 | 0.08 | -0.02 | 5.11 | 70.30 | 24.59 |


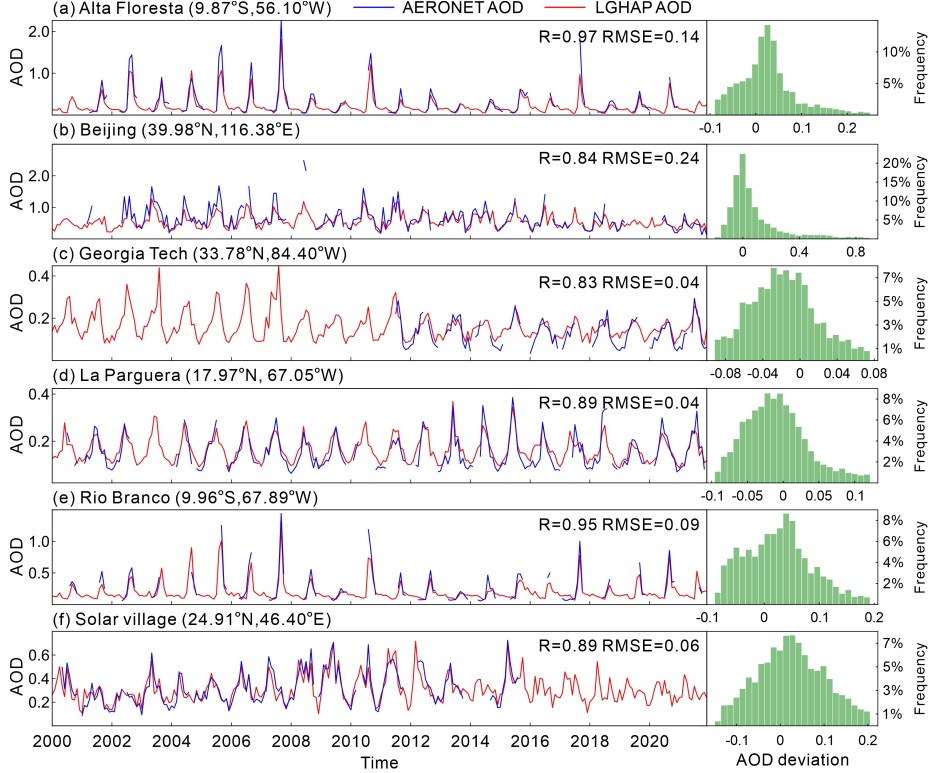


**Figure 5.** Temporal variations in monthly AOD over six AERONET sites with long-term AOD observations during 2000–2021. Panels on
the right present histograms of AOD deviations between LGHAP v2 and AERONET observations at each individual site.

## 4.3 Data accuracy of global gap-free PM$_{2.5}$ concentrations

Global gap-free PM$_{2.5}$ concentration estimates were then derived from gap-filled AOD images by taking advantage of the
novel SeGAT model that was specifically developed to fulfil global PM$_{2.5}$ concentration mapping. More details related to the
performance evaluation of the SCAGAT model were described in another companion study and we hereby focused on the data
accuracy of global gap-free PM$_{2.5}$ concentration estimates. Figure 6 presents the validation accuracy of daily gap-free PM$_{2.5}$
concentration estimates by comparing against ground-based PM$_{2.5}$ concentration records measured at 350 independent (priorly
held-out) monitoring sites. The results indicated that PM$_{2.5}$ concentration estimates derived from the SCAGAT model have
better agreements with ground measured PM$_{2.5}$ concentrations across the globe (R=0.91 and RMSE=9.587 μg m$^{-3}$),
outperforming our traditional PM$_{2.5}$ prediction models without accounting for spatial representativeness of prediction models
during the spatial extrapolation (Bai et al., 2019, 2022a, 2023). As shown in Figure 6e, by taking advantage of the SCAGAT
model, PM$_{2.5}$ concentration estimates over China in LGHAP v2 have a higher data accuracy (R=0.97, RMSE=7.93 μg m$^{-3}$)
than those in LGHAP v1 (R=0.95, RMSE=12.03 μg m$^{-3}$), neglecting different number of validation samples. The data accuracy
was further improved by correcting modelling biases using sparsely distributed in-situ PM$_{2.5}$ concentration measurements via
optimal interpolation, with R improved to 0.95 and RMSE reduced down to 5.7 μg m$^{-3}$. Figs. 6c–6d present site-based
distribution of R and RMSE for LGHAP v2 PM$_{2.5}$ concentration over each individual validation site. Compared to United
States and Europe, as shown in Figures. 6e–6g, larger PM$_{2.5}$ concentration biases were more likely to be observed in Asia
given higher PM$_{2.5}$ loadings therein.

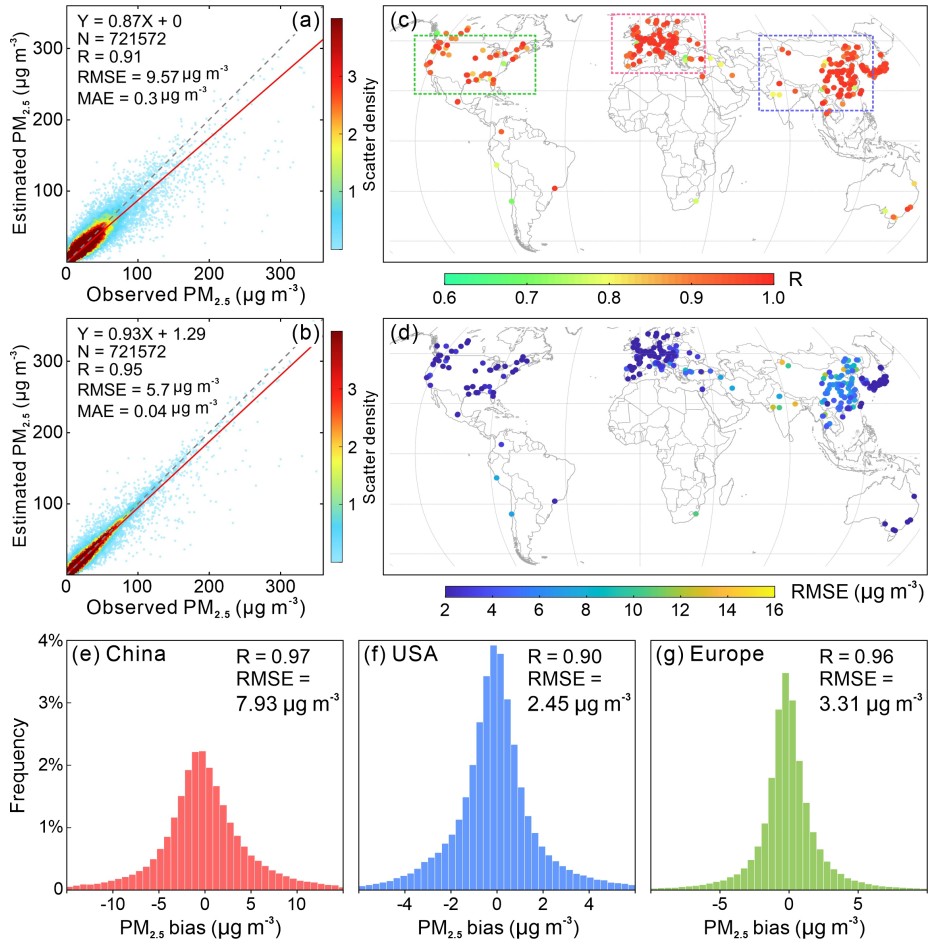

**Figure 6.** Site-based validation accuracy of PM$_{2.5}$ concentration estimates derived from gap-free AOD images using the proposed SeGAT method. (a) Scatter plots between PM$_{2.5}$ estimates derived from the SeGAT model and ground-based PM$_{2.5}$ concentration measurements. (b) Same as Fig. 6a but for gap-free PM$_{2.5}$ estimates fusing ground measured PM$_{2.5}$ concentration from other sites. (c–d) Site-based correlation coefficient I and RMSE for LGHAP v2 PM$_{2.5}$ concentration, respectively. (e–g) Histograms of LGHAP v2 PM$_{2.5}$ concentration bias over China, United States, and Europe, respectively. Note ground-based PM$_{2.5}$ concentration data used here for validation were held out priorly and used neither in model training nor data fusion procedures.

Table 3 presents data accuracy of gap-free PM$_{2.5}$ concentration in LGHAP v2 dataset during the period of 2000−2021 over nations with adequate ground-based measurements of PM$_{2.5}$ concentration records. It indicates that the data accuracy of PM$_{2.5}$ concentration estimates varied across regions, with R changing from 0.71 to 0.98 and RMSE ranging between 1.15 and 32.69 μg m$^{-3}$. Regardless of substantial differences in total number of data pairs across regions, larger RMSEs are mainly observed in regions like Mongolia (32.69 μg m$^{-3}$) and India (25.34 μg m$^{-3}$) where often suffered from high PM$_{2.5}$ loadings. The spatially varying accuracy metrics between regions not only highlight the great complexity in large-scale PM$_{2.5}$ modeling but underscore the critical importance of confirming spatial representativeness via data-driven models, when applying models over other regions for data extrapolation.



**Table 3.** Data accuracy of gap-free $PM_{2.5}$ concentrations in LGHAP v2 dataset by comparing against ground-based $PM_{2.5}$ concentration data in countries with adequate $PM_{2.5}$ concentration measurements. N denotes the total number of $PM_{2.5}$ concentration data pairs for calculating R, RMSE and bias.

| Country | N | R | RMSE ($\mu g\ m^{-3}$) | Bias ($\mu g\ m^{-3}$) | Country | N | R | RMSE ($\mu g\ m^{-3}$) | Bias ($\mu g\ m^{-3}$) |
|---|---|---|---|---|---|---|---|---|---|
| China | 3113160 | 0.97 | 8.27 | 0.36 | Iran | 67434 | 0.74 | 10.14 | −0.09 |
| USA | 2048983 | 0.84 | 3.34 | 0.06 | Brazil | 50252 | 0.81 | 5.63 | 0.78 |
| Japan | 1810436 | 0.96 | 1.82 | 0.07 | Portugal | 47782 | 0.82 | 3.49 | 0.14 |
| Canada | 1206176 | 0.89 | 2.12 | 0.05 | Hungary | 41524 | 0.92 | 4.59 | −0.17 |
| Korea | 526138 | 0.96 | 3.49 | 0.16 | Sweden | 40839 | 0.91 | 1.61 | −0.23 |
| France | 502555 | 0.96 | 2.25 | 0.13 | Norway | 40001 | 0.86 | 2.45 | −0.07 |
| Germany | 472103 | 0.97 | 1.94 | 0.04 | Finland | 38884 | 0.93 | 1.15 | −0.08 |
| Italy | 371888 | 0.93 | 5.23 | 0.04 | South Africa | 35314 | 0.71 | 10.84 | −2.91 |
| UK | 309181 | 0.94 | 1.95 | 0.11 | Serbia | 34795 | 0.87 | 9.70 | 0.01 |
| Spain | 297202 | 0.87 | 2.63 | 0.23 | New Zealand | 26654 | 0.73 | 3.63 | 0.20 |
| Czech | 209274 | 0.97 | 3.38 | 0.24 | Colombia | 26332 | 0.95 | 4.60 | 0.45 |
| Australia | 208772 | 0.72 | 3.70 | −0.03 | Ukraine | 22692 | 0.84 | 5.79 | −0.08 |
| India | 207974 | 0.92 | 25.34 | 1.64 | Bosnia-Herzegovina | 20297 | 0.94 | 12.08 | 1.59 |
| Belgium | 177036 | 0.98 | 1.54 | 0.01 | Greece | 19410 | 0.79 | 5.41 | −0.10 |
| Poland | 175782 | 0.95 | 5.03 | 0.52 | Croatia | 17926 | 0.90 | 5.82 | −0.44 |
| Turkey | 171381 | 0.84 | 10.27 | −0.99 | Switzerland | 14719 | 0.75 | 3.98 | −2.26 |
| Austria | 131186 | 0.97 | 2.28 | −0.14 | Russia | 14357 | 0.84 | 4.06 | 0.58 |
| Netherlands | 119047 | 0.97 | 1.72 | −0.07 | Estonia | 13793 | 0.91 | 1.48 | 0.19 |
| Mexico | 112379 | 0.80 | 11.42 | 0.45 | Lithuania | 13405 | 0.87 | 4.49 | 0.07 |
| Chile | 111416 | 0.80 | 12.64 | 0.16 | Ecuador | 12517 | 0.88 | 2.92 | 0.28 |
| Slovakia | 104892 | 0.95 | 3.77 | 0.18 | Vietnam | 12480 | 0.78 | 12.94 | 0.63 |
| Thailand | 82206 | 0.89 | 13.21 | 1.25 | Macedonia | 10416 | 0.92 | 10.81 | 2.17 |
| Israel | 68012 | 0.83 | 5.08 | 0.32 | Mongolia | 9926 | 0.91 | 32.69 | −0.17 |

In Figure 7, we examined long-term variations in $PM_{2.5}$ concentration in four different cities during 2000–2021. Compared to discrete $PM_{2.5}$ concentration records measured by ground monitors, LGHAP v2 $PM_{2.5}$ concentration time series enabled us to examine long-term variability of haze pollutions across the globe given the gap-free merit. Also, the good agreements between LGHAP v2 $PM_{2.5}$ concentration time series and the unseen (priorly held-out) ground-based $PM_{2.5}$ concentration measurements affirm the high accuracy of LGHAP v2 $PM_{2.5}$ concentration dataset. Therefore, this gap-free $PM_{2.5}$ concentration dataset can be used with high confidence when assessing long-term trends of haze pollution across the globe. As shown, declining trends in $PM_{2.5}$ concentration were observed as early as in 2006 in New York (US), whereas apparent reductions were observed mainly after 2012 in Jilin (China) and 2015 in Toyama (Japan).

Figure 8 presents temporal variations in global annual mean $PM_{2.5}$ concentration from 2000 to 2021. First of all, the daily gap-free merit of LGHAP dataset can seamlessly support the derivation of comparable annual mean $PM_{2.5}$ concentration maps between years as data gap related biases were eliminated due to the usage of daily gap-free $PM_{2.5}$ concentration data. On the other hand, quality-assured annual mean $PM_{2.5}$ concentration maps enable us not only to pinpoint hotspot regions suffering from severe haze pollution but also to examine long-term variability of $PM_{2.5}$ concentrations across the globe. As shown, Mongolia, north India, eastern China, and central Africa were four major regions with relatively high $PM_{2.5}$ loadings. Substantial $PM_{2.5}$ reductions were observed in eastern China since 2014, with $PM_{2.5}$ concentration reduced to a level even comparable to countries in central Asia, and north India was in turn the hotspot region suffering from severer $PM_{2.5}$ pollutions

on the planet.

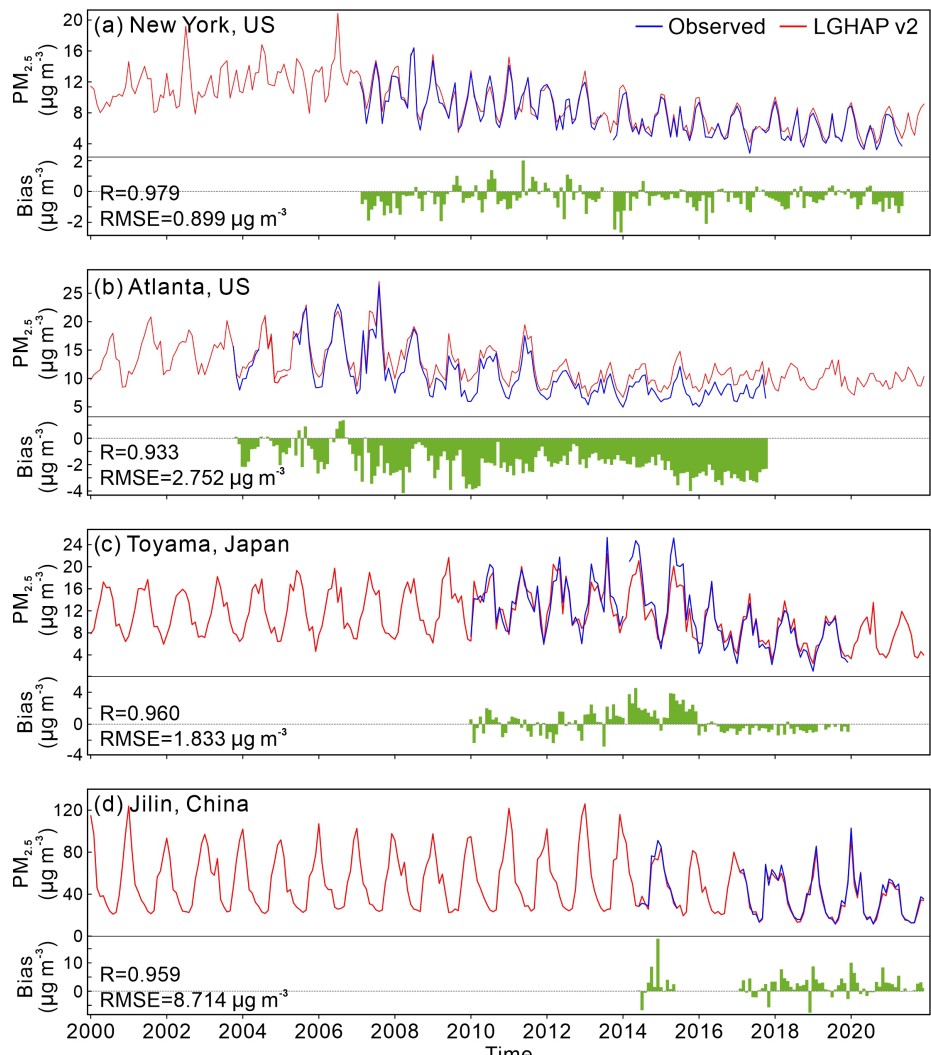

**Figure 7.** An inter-comparison of temporal variations in monthly PM$_{2.5}$ concentration in four different cities between LGHAP v2 and

collocated ground-based PM$_{2.5}$ measurements during 2000−2021.

## 5. Discussion

Spatially contiguous AOD and PM$_{2.5}$ concentration grids are pivotal to regional air quality management, haze pollution

exposure risk assessment, and aerosol radiative forcing diagnosis. By seamlessly gearing up state-of-the-art machine learning

and tensor completion methods, a novel framework of big earth data analytics was developed to fulfil the generation of long-

term high-resolution AOD and PM$_{2.5}$ concentration grids as of 2000 in China (LGHAP v1) in our previous study (Bai et al.,

2022a). Multimodal AODs and related air quality measurements from diverse satellites, numerical models, and ground

monitoring stations were firstly harmonized using random forest-based data-driven models. Multisource AOD data flows were

then weaved neatly as the tensor inputs, from which data gaps in daily MODIS AOD imageries were properly reconstructed



via tensor completion. Finally, gap-free PM$_{2.5}$ concentration grids were mapped from gap-filled AODs using random forest
through machine-learned regression models. This big data analytics framework provided an effective solution to integrate
multimodal earth observations from distinct sources to generate high-quality data products, and the good data accuracies of
these two gap-free datasets also well demonstrated the efficacy of this framework.

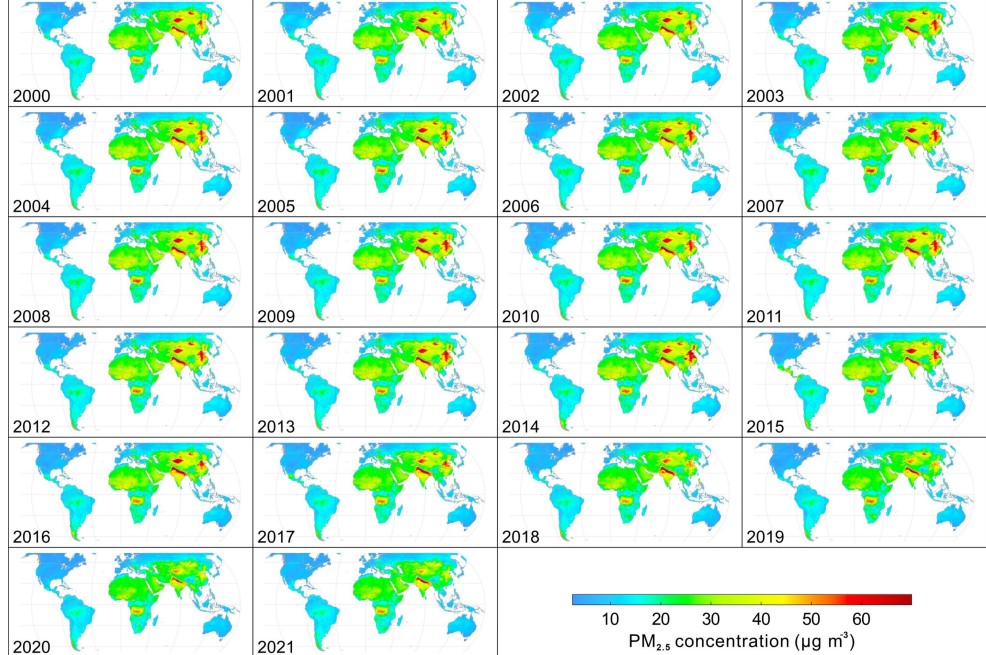

**Figure 8.** Spatial distribution of global annual mean PM$_{2.5}$ concentration derived using LGHAP v2 dataset from 2000 to 2021.
In this study, the big earth data analytics framework proposed in our previous study was adopted to generate global gap-
free AOD and PM$_{2.5}$ concentration grids, i.e., the LGHAP v2 dataset. Despite similar data manipulation procedures, several
new algorithmic enhancement modules were implemented to accommodate the rocketing data size and global scale modeling
demand, not only to improve the computing efficiency but also to reduce modeling biases. Likewise, HOSVD was applied as
the core method for tensor completion to fulfil AOD gap-filling. Nonetheless, previous results indicated a potential drawback
as an equal weight of each data slice in AOD data cube rendered the reconstructed fields more likely to resemble principal
modes determined by HOSVD, and the unique AOD distribution on the target date might be poorly reconstructed, especially
with imbalanced data inputs. To account for this drawback, inspired by widely used attention mechanisms in deep learning
models, we introduced an attention mechanism to weight each data slice in the input tensor, with larger weights assigned to
data better resembling AOD distribution on the target date with more valid observations. In such a research context, spatial
coverage of valid observations in each soft data and mutual information between target and soft data were used as two relevant
metrics to help determine weight assigned to each data slice. A weighted AOD tensor was then calculated and used as the input
tensor to compel tensor completion focusing on data slices more similar to the target image rather than all available data. As
demonstrated by the ablation experiments shown in Figure 2, AOD fields reconstructed from the attention-reinforced tensor
better resembled actual AOD distributions in the target AOD$_{Terra}$ images than those derived from raw AOD tensor without
applying attention mechanism.
Meanwhile, an adaptive background field updating scheme was introduced to update prior information in the target
AOD$_{Terra}$ images during each iteration of tensor decomposition and reconstruction, and the ultimate goal was to mitigate the
influence of prior information on the reconstruction accuracy, particularly reducing the probability of possible propagation of
large modelling biases in AOD$_{M2}$ to the reconstructed AOD fields. Compared to invariant prior information, adaptively updated
prior information enabled us to not only improve the reconstruction efficiency but also significantly reduce the probability of
large error propagation from numerical AOD simulations. Despite these algorithmic improvements, the inter-comparison
results even indicated a slightly reduced data accuracy of gap-filled AODs in China compared to those in LGHAP v1 dataset.
Further investigations revealed this was mainly due to a relatively poor data accuracy of AOD$_{M2}$ data since a global-scale
rather than regional downscaling model was applied to harmonize AOD$_{M2}$ in China. This in turn underscores the vital
importance of data cleaning procedures on reducing bias levels of each supplementary data to manage the total error budget in
the final analyzed data fields when performing big data analytics.
As illustrated in Figure 9, gap-filled AOD grids with a daily 1-km resolution enable us to better monitor global aerosol
distribution and variations in space and time. Aerosol related environmental disturbance episodes such as sandstorm, wildfire,
and haze pollution events can be well captured by rising AODs at the regional scale. Most critically, the gap-filled AOD dataset
provides us an unprecedent opportunity to monitor aerosol loadings and variations even under cloud covers, e.g. haze pollution
episodes over southern India and eastern China shown in Figures 9d and 9e, largely benefiting from the intelligent
spatiotemporal pattern recognition and learning as well as the assimilation of air quality measurements from ground monitoring
stations and numerical aerosol diagnostics. While such a global air quality mapping approach greatly facilitates the surveillance
and management of air pollution around the world, the high-resolution gap-free AOD and PM$_{2.5}$ concentration dataset would
also largely reduce the uncertainty in health-related aerosol exposure risk assessment.

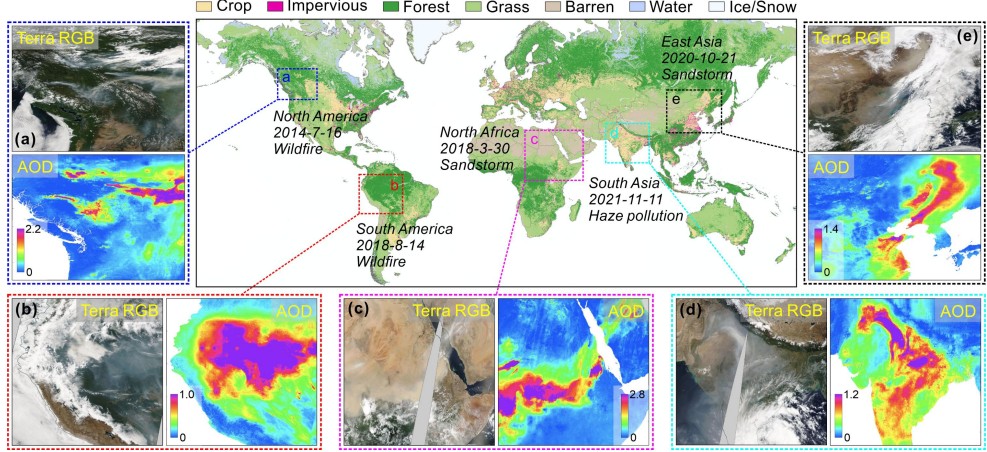

**Figure 9.** An illustration of AOD responses to wild fire, sand storm, and haze pollution episodes across the globe as characterized by gap-
free AOD in LGHAP v2 dataset. Global map in the middle panel shows spatial distribution of major land cover types in 2020.

By taking advantage of the LGHAP v2 AOD dataset, global AOD variation trends were carefully examined. Fig. 10a
presents AOD deviations between AOD averages during the first and the second decade across the globe. As shown, substantial
AOD increases in the 21st century present primarily over India (G) and central Africa (I), with remarkable AOD decreases
observed in the middle of South America. In North America, AOD increases were mainly observed in Canada and western US
(A) whereas AOD decreases were found in eastern US (B). Also, referring to temporally varied AOD trends in regions A and
B, we may observe evident AOD increasing trends in US since 2012, while the significant decreasing trends in eastern US
were even totally reversed after 2015. This effect could be partially linked to more frequent and intensive wildfire emissions
in the second decade of 2000s in north America (Burke et al., 2023; Wei et al., 2021b). Similar effect was also observed in
Europe (C), with an apparent slowdown in AOD decreasing trend after 2010.

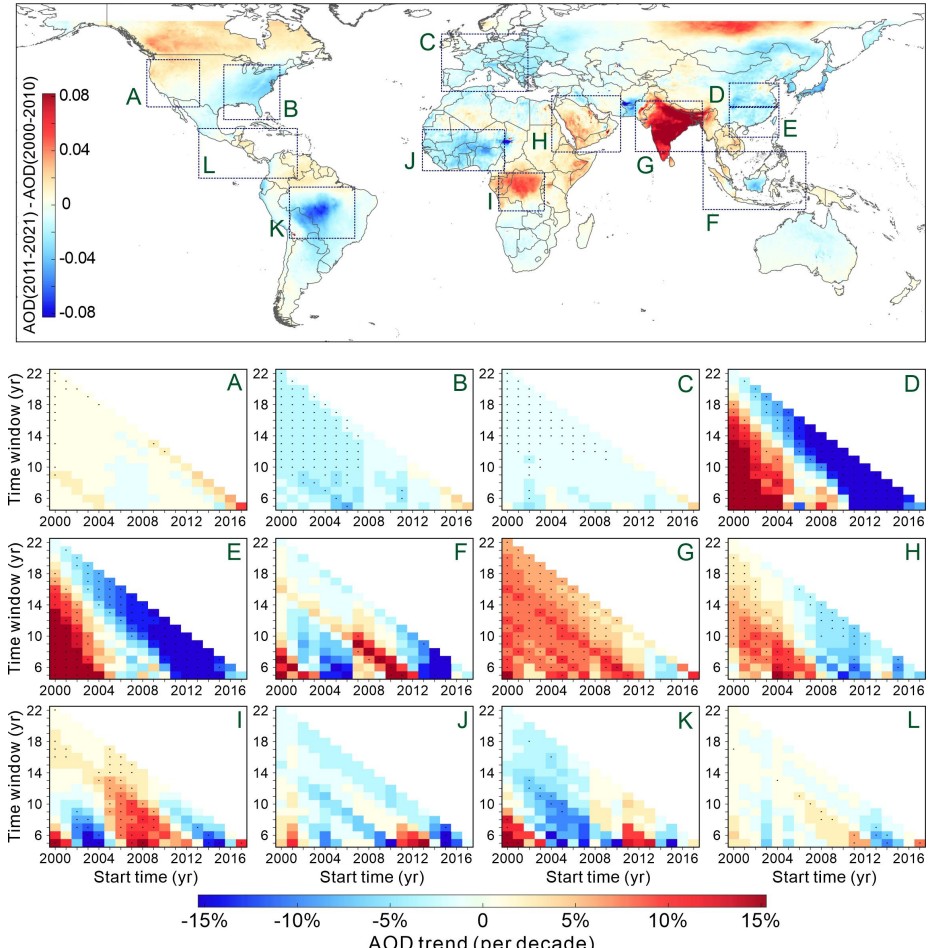


**Figure 10.** AOD trends over twelve regions of interest across the globe from 2000 to 2021 estimated from gap-free AOD in LGHAP v2
dataset. The top panel shows spatial distribution of global AOD deviations between the first and second decade in 2000s. Twelve diagrams
in the bottom panel show the linear trend of mean AOD over the outlined region of interest at different starting time with varying sizes of
time window.

Apparent inverse effects were also observed in China but with totally different temporal transition patterns. As shown,

statistically significant AOD increasing trends were observed in eastern (D) and southern (E) China in the first decade, whereas
increasing trends started to slow down since 2007 and a sudden reverse to decreasing trends was observed after 2010. More
importantly, this was also the most significant AOD decreasing trend in 2010s around the world. These observational evidences
affirm the great success of clean air actions in improving air quality in China during the past decades (Bai et al., 2022a; Liang
et al., 2020; Zhang et al., 2019). Similar temporal variation pattern was also observed in Middle East (H) but with relatively
weak trends. In contrast, India (G) was the hotspot area showing an increasing trend in AOD throughout the 2000s, despite a
short period of increasing hiatus during 2013–2015.

In this study, global gap-free $PM_{2.5}$ concentrations were derived on the basis of gap-filled AOD grids by taking advantage

of a novel SCAGAT deep learning model, which was specifically developed to fulfil global scale $PM_{2.5}$ concentration mapping.



Differing from many other modeling practices, spatial representativeness of data-driven models was accounted for by
SCAGAT, providing a unique solution to model $PM_{2.5}$ concentration over regions even without $PM_{2.5}$ monitoring sites. The
availability of daily gap-free $PM_{2.5}$ concentration grids also favor the assessment of pandemic impacts on regional air quality.
Figs. 11a and 11b in the middle panel present spatial distribution of $PM_{2.5}$ concentration before and during the COVID-19
pandemic, respectively. Neglecting long-term variation trends in $PM_{2.5}$ concentration, the substantial $PM_{2.5}$ decreases in the
middle and eastern China as well as central Europe clearly indicate the positive effect of pandemic related mobility restrictions
on air quality improvement, by comparing $PM_{2.5}$ concentration in 2019 and 2020 during the synchronous period. In contrast,
$PM_{2.5}$ reductions were relatively small in US due to the lack of mobility restriction measures, with apparent $PM_{2.5}$ reductions
observed mainly in Chicago. Overall, the availability of LGHAP v2 dataset enables us to better investigate global aerosol
variations and to assess $PM_{2.5}$ related health risk via exposure assessment.

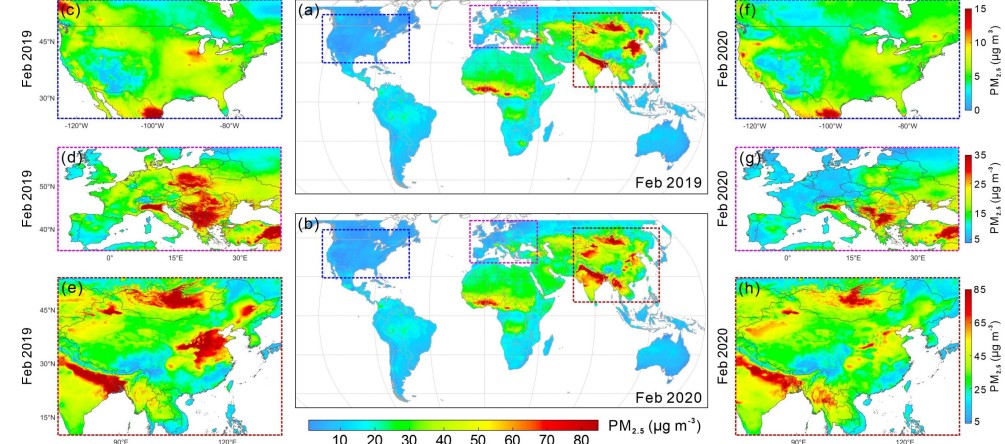


**Figure 11.** Impacts of COVID-19 pandemic on $PM_{2.5}$ concentrations in United States, Europe, and China. $PM_{2.5}$ concentrations from LGHAP
v2 were averaged over the synchronous period in 2019 and 2020 for inter-comparison.
**6. Data Availability**
The LGHAP v2 dataset provides global gap-free AOD and $PM_{2.5}$ concentration grids from 2000 to 2021 with daily 1-km
resolution. To facilitate data sharing, each daily map was saved as one NetCDF file, and data in each individual month was
then archived as a zip file. Due to the data storage limitations, data in one year were archived as one single dataset. Table 4
provides the permanent digital object identifiers for each individual dataset. All datasets were available at the LGHAP
community link via https://zenodo.org/communities/ecnu_lghap (Bai et al., 2023a). Data user guide and visualization codes
(Python, MATLAB, R, and IDL) were also provided to guide the users to retrieve data from the NetCDF files, which can be
accessible at https://doi.org/10.5281/zenodo.10216396.
**7. Conclusion**
In this study, the LGHAP v2 dataset, a heritage of LGHAP which provides long-term gap-free AOD and PM concentration
grids with daily 1-km resolution in China, was generated to provide gap-free AOD and $PM_{2.5}$ concentration grids with the
same resolution as of 2000 across the globe, by taking advantage of an improved big earth data analytics approach. Ground
validation results demonstrate high accuracies of these two gap-free products, with AOD having a correlation of 0.85 and



RMSE of 0.14 compared to AERONET AOD observations, slightly worse than the original MCD19A2 product (R=0.88 and
RMSE=0.11). Site-based validation results also indicate that $PM_{2.5}$ concentration estimates derived from gap-free AOD via
SCAGAT show a good agreement with held-out ground-based $PM_{2.5}$ measurements, with R of 0.91 and RMSE of 9.57 μg m$^{-3}$,
and the data accuracy was further improved to 0.95 and 5.7 μg m$^{-3}$ with the fusion of ground $PM_{2.5}$ measurements. To our
knowledge, this is the first two-decade-long global gap-free AOD and $PM_{2.5}$ concentration dataset with such a high resolution.
**Table 4.** List of data links for AOD and $PM_{2.5}$ concentration grids in LGHAP v2 dataset in each individual year of 2000–2021.

| Year | LGHAP v2 AOD grids | LGHAP v2 $PM_{2.5}$ grids |
|---|---|---|
| 2000 | https://doi.org/10.5281/zenodo.8281206 | https://doi.org/10.5281/zenodo.8307595 |
| 2001 | https://doi.org/10.5281/zenodo.8281216 | https://doi.org/10.5281/zenodo.8307597 |
| 2002 | https://doi.org/10.5281/zenodo.8281218 | https://doi.org/10.5281/zenodo.8307599 |
| 2003 | https://doi.org/10.5281/zenodo.8281222 | https://doi.org/10.5281/zenodo.8307601 |
| 2004 | https://doi.org/10.5281/zenodo.8281226 | https://doi.org/10.5281/zenodo.8307605 |
| 2005 | https://doi.org/10.5281/zenodo.8281228 | https://doi.org/10.5281/zenodo.8307607 |
| 2006 | https://doi.org/10.5281/zenodo.8287125 | https://doi.org/10.5281/zenodo.8308225 |
| 2007 | https://doi.org/10.5281/zenodo.8287129 | https://doi.org/10.5281/zenodo.8308227 |
| 2008 | https://doi.org/10.5281/zenodo.8287133 | https://doi.org/10.5281/zenodo.8308231 |
| 2009 | https://doi.org/10.5281/zenodo.8287995 | https://doi.org/10.5281/zenodo.8308233 |
| 2010 | https://doi.org/10.5281/zenodo.8288389 | https://doi.org/10.5281/zenodo.8308237 |
| 2011 | https://doi.org/10.5281/zenodo.8288395 | https://doi.org/10.5281/zenodo.8310586 |
| 2012 | https://doi.org/10.5281/zenodo.8288397 | https://doi.org/10.5281/zenodo.8310590 |
| 2013 | https://doi.org/10.5281/zenodo.8287207 | https://doi.org/10.5281/zenodo.8310702 |
| 2014 | https://doi.org/10.5281/zenodo.8288387 | https://doi.org/10.5281/zenodo.8310704 |
| 2015 | https://doi.org/10.5281/zenodo.8289613 | https://doi.org/10.5281/zenodo.8310706 |
| 2016 | https://doi.org/10.5281/zenodo.8289615 | https://doi.org/10.5281/zenodo.8310708 |
| 2017 | https://doi.org/10.5281/zenodo.8294100 | https://doi.org/10.5281/zenodo.8310711 |
| 2018 | https://doi.org/10.5281/zenodo.8301364 | https://doi.org/10.5281/zenodo.8313603 |
| 2019 | https://doi.org/10.5281/zenodo.8301367 | https://doi.org/10.5281/zenodo.8313611 |
| 2020 | https://doi.org/10.5281/zenodo.8301375 | https://doi.org/10.5281/zenodo.8313613 |
| 2021 | https://doi.org/10.5281/zenodo.8301379 | https://doi.org/10.5281/zenodo.8313615 |

Data gaps in satellite-based AOD images were filled using a similar big data analytics approach as used to generate the
LGHAP dataset in China but with several new algorithmic improvements. The ablation experiments well demonstrated the
effectiveness and advantages of applying attention mechanism to weight each slice of soft data in AOD tensor during the tensor
completion procedure. Also, updating prior information in the target image after each iteration not only helps mitigate the
probability of error propagation from numerical aerosol diagnostics to the final reconstructed field but also improves the
convergence speed of tensor completion. Moreover, this study provides a good illustration of big earth data analytics to
generate high-quality datasets by synergistically integrating and assimilating multimodal data from diverse sources via
machine learning. The last but not least, this big data analytics approach can be also used to fulfil near-term gap-free AOD
mapping by simply replacing aerosol reanalysis with numerical AOD forecasts (e.g., CAMS AOD forecasts).
This study also provides new insights on how to deal with the scaling effect when establishing large scale $PM_{2.5}$ prediction
models. Rather than creating a global model by gathering all paired data into one training set, site-specific $PM_{2.5}$ prediction



models were firstly established using random forest, and a graph attention network was then applied to establish a spatial
interpolation model on the basis of PM$_{2.5}$ estimates derived from random forest models trained over sites with similar scene
features as the target grid. Since there is no need to establish regional estimation models, such a philosophy not only improves
the modeling accuracy but also solves the scaling problem in large scale modeling practices.
The LGHAP v2 dataset is publicly accessible from the links given above. Given the gap-free and high-resolution merit,
this dataset can be used to deepen our understanding of aerosol climatic effects as well as PM$_{2.5}$ exposure risks and related
health outcomes at the global scale. Also, the researchers are encouraged to use this dataset to better evaluate the sustainable
development goals related to urban air quality across the globe.
**Competing interests**
The contact author has declared that none of the authors has any competing interests.
**Acknowledgments**
This study was supported by the National Natural Science Foundation of China (Grant No. 42171309), the International
Research Center of Big Data for Sustainable Development Goals (Grant No. CBAS2022GSP07), the Foreign Technical
Cooperation and Scientific Research Program (Grant No. E3KZ0301), and the Director's Fund of Key Laboratory of
Geographic Information Science (Ministry of Education), East China Normal University (Grant No. KLGIS2023C01). The
authors would like to express gratitude to relevant organizations and data archive services for their great efforts in providing
essential data sources used in this study to support the generation of global LGHAP v2 dataset.

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
