# Peer review of "LGHAP v2: A global gap-free aerosol optical depth and PM2.5 concentration dataset since 2000 derived via big Earth data analytics"

_Earth System Science Data, 2023_

## Author Response (AR1)

**Reply to reviewer #1's comments**

In this study, the authors introduced a new aerosol dataset, LGHAP v2, i.e., a global gap-free AOD and PM$_{2.5}$ concentration dataset with daily and 1-km resolution covering the period of 2000–2021 using an improved big data analytics approach. This new dataset is an extended version of LGHAP which provides gap-free AOD and PM concentrations in China. These gap-free datasets provide essential high-quality data sources to advance our understanding of global variations of aerosol and haze pollutions, greatly facilitating aerosol radiative forcing estimation and human exposure risk assessment. Overall, the paper is well organized, the dataset is publicly accessible and the data accuracy is fairly evaluated. The following comments could be considered to further improve the manuscript.

Reply: Thank you so much for your positive responses. In this revision, comments and suggestions from you and another reviewer have been carefully considered and addressed, which greatly help us to enhance the clarity and quality of this paper. Below gives our point-by-point response to your comments.

1. Section 2.2.2: how about the temporal resolution of ground-based PM$_{2.5}$ concentrations, hourly data during the satellite overpass or daily average? Needs to clarify.

Reply: Thanks!! The ground-based monitoring stations used in this study were operated independently by different countries and/or organizations. To ensure uniformity and comparability of these data, we conducted rigorous preprocessing and converted time-series data to a daily average scale. We have clarified this point in this revision to avoid the potential readership gap.

2. Ground-based PM$_{2.5}$ concentration measurements were collected from different agencies using different instruments, so data heterogeneity is a concern and the authors should discuss this issue in the manuscript.

Reply: Thank you for your constructive suggestion! We have provided more relevant discussions on this issue in the revised manuscript.

3. How about the data in 2022 and 2023, will the authors release data in these two

years publicly?

**Reply**: Yes! Gap-free data in this two-year will be released publicly soon once the computation is finished. All these data are available for public access.

4. Is it possible to implement the proposed method for near-real-time data generation? What are potential limitations?

**Reply**: Thanks!! We intend to implement the proposed method for near-term operation. However, there are still some challenges to address, including not only the availability and latency of input data sources but also huge computational resources requirement for data processing, and the frequency of model updating. Also, factors such as data transmission efficiency may also impact the effectiveness of NRT data production. We have discussed this point in the revised manuscript as well.

5. Many abbreviations were used without providing the full name, e.g., MODIS, R, RMSE in the abstract, please check.

**Reply**: Thanks! All these issues have been well addressed in this revision.

6. Line 24: "multi-source" should be multisource, no hyphen. Similarly, "data base" in line 38 should be database.

**Reply**: Thanks!! These grammar issues have been carefully checked and corrected.

**Reply to reviewer #2's comments**

LGHAP v2, a global gap-free aerosol optical depth and PM$_{2.5}$ concentration dataset, was hereby introduced. As a heritage of the LGHAP published earlier in 2022, this new dataset not only has a global coverage but with an enhanced data accuracy, largely benefiting from several new algorithmic improvements. Overall, the paper is well written, with clear descriptions of algorithms and validation results of this new dataset. The following gives my personal concerns that could be addressed to improve this paper.

**Reply**: Thanks for your positive feedback! We have revised the manuscript carefully by considering your valuable comments and suggestions.

1. There are many abbreviations used without clearly spelling out the full name upon their first uses. Please ensure that the full name of each term should come first, followed by the abbreviation in parentheses. Please check this issue and make revisions throughout the paper.

**Reply**: Thanks for pointing out! We have carefully revised the manuscript to ensure that all abbreviations are accompanied by their full names upon the first usage.

2. There are many lengthy sentences, e.g., lines 105–109, 169–173, try to break them down into several short ones to ease the readership.

**Reply**: Thanks for your valuable suggestion! These lengthy sentences have been rewritten in this revision to improve the readability.

3. Try to move Table 1 to the place right beneath its call out, the current place is too far away.

**Reply**: Thanks!! Table 1 have been moved to the front part right beneath its call out.

4. Line 220-221: "while emphasizing the newly developed algorithmic components in the following.", what does the "following" refer to, the subsections below or any other sections? Needs to clarify.

**Reply**: Thank you for pointing out! The term "following" refers specifically to the subsection "3.1.2 Algorithmic improvements". We have reworded the sentence to ease the readability.

5. Equation 3 and line 276: why did the authors choose to use mutual information? How about correlation coefficients, any explanations?

**Reply**: Thanks for your evaluable and constructive question! In this study, mutual information was chosen for its effectiveness in measuring mutual dependency between target $AOD_{Terra}$ imagery and historical AOD imageries. Unlike correlation coefficients, which primarily capture linear relationships, mutual information is adept at capturing nonlinear correlations, which can be particularly useful for screening historical similar AOD imageries. Additionally, the design of our updated tensor-flow-based AOD reconstruction module can effectively leverage the uncorrelated relationship between imageries, thereby providing more prior information for filling data gaps. In the revision, we have included additional explanations for a comprehensive understanding.

6. Line 297: "In such a context, large modeling biases in $AOD_{M2}$ might be introduced into the final reconstructed fields", is there any evidence to support this claim?

**Reply**: Thanks!! Our claims regarding the potential bias introduction to the final reconstructed fields are associated with the methodology proposed in our previous study, namely, the benchmark method mentioned in this study. 5% random samples from the downscaled MERRA-2 AOD imagery (referred to as $AOD_{M2}$) were used as the prior information to initiate AOD gap-filling. While this approach facilitated the improvement of convergence speed during tensor completion, it also raises the problem that the spatial pattern of the reconstructed fields over regions with excessive data gaps is more likely to resemble the distribution of $AOD_{M2}$. This effect was primarily due to the equal weight assigned to both observational AOD and $AOD_{M2}$ in the tensor completion process, resulting in a relative weaker influence of sparse observational AOD data derived from air quality measurements and other satellite observations. Consequently, any large modeling bias present in $AOD_{M2}$ could be introduced into the final reconstructed result, as shown in Figure S5 (the third row). However, with the implementation of the adaptive prior information updating strategy, this defect was largely addressed (shown in the fourth row). This scheme allows for an adaptive adjustment of background information based on available observational data, thereby mitigating the influence of modeling biases in $AOD_{M2}$ on the final reconstructed fields.

[Figure]

**Figure S5**. Performance evaluation of the adaptive background information updating module on improving AOD reconstruction patterns. Intercomparisons were conducted between the benchmark method (the method developed in Bai et al. (2022) to generate LGHAP dataset in China) and the one embedding adaptive background information updating module.

7. Figure 2c: it seems the algorithmic improvements contribute little to the reconstruction accuracy.

**Reply**: Thanks!! Figure 2c shown here is to elaborate the truth that our algorithmic improvements still have limitations in improving the reconstruction accuracy. Specifically, similar as the benchmark method, the beneficial effects of algorithmic improvements could be largely cancelled when input imageries have extensive data gaps. This could be attributable to limited observational data in the target imagery for reference to pinpoint historical imageries with similar pattern. In other words, the substantial data scarcity greatly reduces the effectiveness of the newly implemented

attention mechanism. In this revision, we have provided more relevant discussions on this issue to bridge the readership gap.

8. Line 454: "in another companion study", please provide the name of the study with a citation support at here for reference, if possible.

Reply: Thanks!! We have provided the name of our companion study along with a formal citation in this revision.

9. Line 613: change "two-decade-long" to twenty-year long.

Reply: Thanks!! It has been corrected per your suggestion.